# MULTI-SKILL MOBILE MANIPULATION FOR OBJECT REARRANGEMENT

**Jiayuan Gu[1], Devendra Singh Chaplot[2], Hao Su[1], Jitendra Malik[2,3]**
[1]UC San Diego, [2]Meta AI Research, [3]UC Berkeley

## ABSTRACT

We study a modular approach to tackle long-horizon mobile manipulation tasks for object rearrangement, which decomposes a full task into a sequence of subtasks. To tackle the entire task, prior work chains multiple stationary manipulation skills with a point-goal navigation skill, which are learned individually on subtasks. Although more effective than monolithic end-to-end RL policies, this framework suffers from compounding errors in skill chaining, *e.g.*, navigating to a bad location where a stationary manipulation skill can not reach its target to manipulate. To this end, we propose that the manipulation skills should include mobility to have flexibility in interacting with the target object from multiple locations and at the same time the navigation skill could have multiple end points which lead to successful manipulation. We operationalize these ideas by implementing mobile manipulation skills rather than stationary ones and training a navigation skill trained with region goal instead of point goal. We evaluate our multi-skill mobile manipulation method **M3** on 3 challenging long-horizon mobile manipulation tasks in the Home Assistant Benchmark (HAB), and show superior performance as compared to the baselines.

## 1  INTRODUCTION

Building AI with embodiment is an important future mission of AI. Object rearrangement (Batra et al., 2020) is considered as a canonical task for embodied AI. The most challenging rearrangement tasks (Szot et al., 2021; Ehsani et al., 2021; Gan et al., 2021) are often long-horizon mobile manipulation tasks, which demand both navigation and manipulation abilities, *e.g.*, to move to certain locations and to pick or place objects. It is challenging to learn a monolithic RL policy for complex long-horizon mobile manipulation tasks, due to challenges such as high sample complexity, complicated reward design, and inefficient exploration. A practical solution to tackle a long-horizon task is to decompose it into a set of subtasks, which are tractable, short-horizon, and compact in state or action spaces. Each subtask can be solved by designing or learning a skill, so that a sequence of skills can be chained to complete the entire task (Lee et al., 2018; Clegg et al., 2018; Lee et al., 2019; 2021). For example, skills for object rearrangement can be picking or placing objects, opening or closing fridges and drawers, moving chairs, navigating in the room, *etc*.

Achieving successful object rearrangement using this modular framework requires careful subtask formulation such that skills trained for these subtasks can be chained together effectively. We define three desirable properties for skills to solve diverse long-horizon tasks: **achievability, composability, and reusability**. Note that we assume each subtask is associated with a set of initial states. Then, *achievability* quantifies the portion of initial states solvable by a skill. A pair of skills are *composable* if the initial states achievable by the succeeding skill can encompass the terminal states of the preceding skill. This encompassment requirement is necessary to ensure robustness to mild compounding errors. However, trivially enlarging the initial set of a subtask increases learning difficulty and may lead to many unachievable initial states for the designed/learned skill. Last, a skill is *reusable* if it can be directly chained without or with limited fine-tuning (Clegg et al., 2018; Lee et al., 2021). According to our experiments, effective subtask formulation is critical though largely overlooked in the literature.

---

[1]Project website: `https://sites.google.com/view/hab-m3`
[2]Codes: `https://github.com/Jiayuan-Gu/hab-mobile-manipulation`

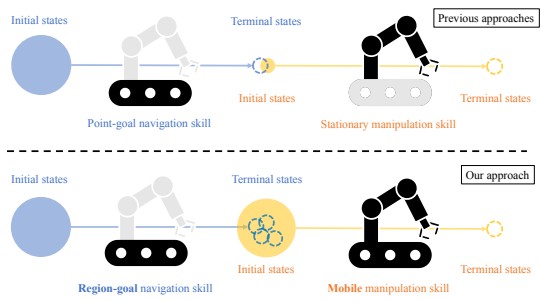
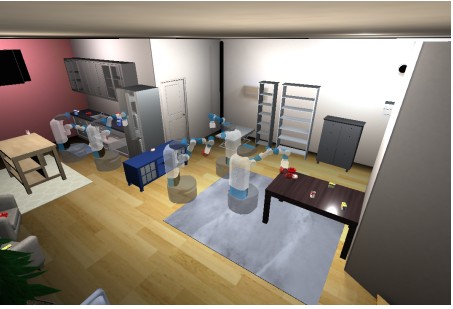

(a) Method Overview        (b) The Home Assistant Benchmark

Figure 1: 1a provides an overview of our multi-skill mobile manipulation (**M3**) method. The inactive part of the robot is colored gray. Previous approaches exclusively activate either the mobile platform or manipulator for each skill, and suffer from compounding errors in skill chaining given limited composability of skills. We introduce mobility to manipulation skills, which effectively enlarges the feasible initial set, and a region-goal navigation reward to facilitate learning the navigation skill. 1b illustrates one task (*SetTable*) in the Home Assistant Benchmark (Szot et al., 2021), where the robot needs to navigate in the room, open the drawers or fridge, pick multiple objects in drawers or fridge and place them on the table. Best viewed in motion at the project website[1].

In the context of mobile manipulation, skill chaining poses many challenges for subtask formulation. For example, an imperfect navigation skill might terminate at a bad location where the target object is out of reach for a stationary manipulation skill (Szot et al., 2021). To tackle such "hand-off" problems, we investigate how to formulate subtasks for mobile manipulation. First, we replace stationary (fixed-base) manipulation skills with mobile counterparts, which allow the base to move when the manipulation is undertaken. We observe that mobile manipulation skills are more robust to compounding errors in skill chaining, and enable the robot to make full use of its embodiment to better accomplish subtasks, *e.g.*, finding a better location with less clutter and fewer obstacles to pick an object. We emphasize how to generate initial states of manipulation skills as a trade-off between *composability* and *achievability* in Sec 4.1. Second, we study how to translate the start of manipulation skills to the navigation reward, which is used to train the navigation skill to connect manipulation skills. Note that the goal position in mobile manipulation plays a very different role from that in point-goal (Wijmans et al., 2019; Kadian et al., 2020) navigation. On the one hand, the position of a target object (*e.g.*, on the table or in the fridge) is often not directly navigable; on the other hand, a navigable position close to the goal position can be infeasible due to kinematic and collision constraints. Besides, there exist multiple feasible starting positions for manipulation skills, yet previous works such as Szot et al. (2021) train the navigation skill to learn a single one, which is selected heuristically and may not be suitable for stationary manipulation. Thanks to the flexibility of our mobile manipulation skills, we devise a region-goal navigation reward to address those issues, detailed in Sec 4.2.

In this work, we present our improved multi-skill mobile manipulation method **M3**, where mobile manipulation skills are chained by the navigation skill trained with our region-goal navigation reward. It achieves an average success rate of 63% on 3 long-horizon mobile manipulation tasks in the Home Assistant Benchmark (Szot et al., 2021), as compared to 50% for our best baseline. Fig 1 provides an overview of our method and tasks. Our contributions are listed as follows:

1. We study how to formulate mobile manipulation skills, and empirically show that they are more robust to compounding errors in skill chaining than stationary counterparts;

2. We devise a region-goal navigation reward for mobile manipulation, which shows better performance and stronger generalizability than the point-goal counterpart in previous works;

3. We show that our improved multi-skill mobile manipulation pipeline can achieve superior performance on long-horizon mobile manipulation tasks without bells and whistles, which can serve as a strong baseline for future study.

## 2 RELATED WORK

### 2.1 MOBILE MANIPULATION

Rearrangement (Batra et al., 2020) is "to bring a given physical environment into a specified state". We refer readers to Batra et al. (2020) for a comprehensive survey. Many existing RL tasks can be considered as instances of rearrangement, *e.g.*, picking and placing rigid objects (Zhu et al., 2020; Yu et al., 2020) or manipulating articulated objects (Urakami et al., 2019; Mu et al., 2021). However, they mainly focus on stationary manipulation (Urakami et al., 2019; Zhu et al., 2020; Yu et al., 2020) or individual, short-horizon skills (Mu et al., 2021). Recently, several benchmarks like Home Assistant Benchmark (HAB) (Szot et al., 2021), ManipulaTHOR (Ehsani et al., 2021) and ThreeDWorld Transport Challenge (Gan et al., 2021), are proposed to study long-horizon mobile manipulation tasks. They usually demand that the robot rearranges household objects in a room, requiring exploration and navigation (Anderson et al., 2018; Chaplot et al., 2020) between interacting with objects entirely based on onboard sensing, without any privileged state or map information.

Mobile manipulation (RAS, 2022) refers to "robotic tasks that require a synergistic combination of navigation and interaction with the environment". It has been studied long in the robotics community. Ni et al. (2021) provides a summary of traditional methods, which usually require perfect knowledge of the environment. One example is task-and-motion-planning (TAMP) (Srivastava et al., 2014; Garrett et al., 2021; 2020). TAMP relies on well-designed state proposals (grasp poses, robot positions, *etc.*) to sample feasible trajectories, which is computationally inefficient and unscalable for complicated scenarios.

Learning-based approaches enable the robot to act according to visual observations. Xia et al. (2021) proposes a hierarchical method for mobile manipulation in iGibson (Xia et al., 2020), which predicts either a high-level base or arm action by RL policies and executes plans generated by motion-planning to achieve the action. However, the arm action space is specially designed for a primitive action *pushing*. Sun et al. (2022) develops a real-world RL framework to collect trash on the floor, with separate navigation and grasping policies. Ehsani et al. (2021); Ni et al. (2021) train an end-to-end RL policy to tackle mobile pick-and-place in ManipulaTHOR (Ehsani et al., 2021). However, the reward function used to train such an end-to-end policy usually demands careful tuning. For example, Ni et al. (2021) shows that a minor modification (a penalty for disturbance avoidance) can lead to a considerable performance drop. The vulnerability of end-to-end RL approaches restricts scalability. Most prior works in both RL and robotics separate mobile the platform and manipulator, to "reduce the difficulty to solve the inverse kinematics problem of a kinematically redundant system" (Sereinig et al., 2020; Sandakalum & Ang Jr, 2022). Wang et al. (2020) trains an end-to-end RL policy based on the object pose and proprioception to simultaneously control the base and arm. It focuses on picking a single object up in simple scenes, while our work addresses long-horizon rearrangement tasks that require multiple skills.

Szot et al. (2021) adopts a different hierarchical approach for mobile manipulation. It uses task-planning (Fikes & Nilsson, 1971) to generate high-level symbolic goals, and individual skills are trained by RL to accomplish those goals. It outperforms the monolithic end-to-end RL policy and the classical sense-plan-act robotic pipeline. It is scalable since skills can be composited to solve different tasks, and benefit from progress in individual skill learning (Yu et al., 2020; Mu et al., 2021). Moreover, different from other benchmarks, the HAB features continuous motor control (base and arm), interaction with articulated objects (opening drawers and fridges), and complicated scene layouts. Thus, we choose the HAB as the platform to study long-horizon mobile manipulation.

### 2.2 SKILL CHAINING FOR LONG-HORIZON TASKS

Szot et al. (2021) observes that sequentially chaining multiple skills suffers from "hand-off" problems, where a preceding skill terminates at a state that the succeeding skill has either never seen during training or is infeasible to solve. Lee et al. (2018) proposes to learn a transition policy to connect primitive skills, but assumes that such a policy can be found through random exploration. Lee et al. (2021) regularizes the terminal state distribution of a skill to be close to the initial set of the following skill, through a reward learned with adversarial training. Most prior skill chaining methods focus on fine-tuning learned skills. In this work, we instead focus on subtask formulation for skill chaining, which directly improves composability and reusability without additional computation.

## 3 PRELIMINARY

### 3.1 HOME ASSISTANT BENCHMARK (HAB)

The Home Assistant Benchmark (HAB) (Szot et al., 2021) includes 3 long-horizon mobile manipulation rearrangement tasks (*TidyHouse, PrepareGroceries, SetTable*) based on the ReplicaCAD dataset, which contains a rich set of 105 indoor scene layouts. For each episode (instance of task), rigid objects from the YCB (Calli et al., 2015) dataset are randomly placed on annotated supporting surfaces of receptacles, to generate clutter in a randomly selected scene. Here we provide a brief description of these tasks.

**TidyHouse**: Move 5 objects from starting positions to goal positions. Objects and goals are located in open receptacles (*e.g.*, table, kitchen counter) rather than containers. Complex scene layouts, diverse receptacles, dense clutter all pose challenges. The task implicitly favors collision-free behavior since a latter target object might be knocked out of reach when a former object is moved by the robot.
**PrepareGroceries**: Move 2 objects from the fridge to the counters and move an object from the counter to the fridge. The fridge is fully open initially. The task requires picking and placing an object in a cluttered receptacle with restricted space.
**SetTable**: Move a bowl from a drawer to a table, and move a fruit from the fridge to the bowl on the table. Both the drawer and fridge are closed initially. The task requires interaction with articulated objects as well as picking objects from containers.

All the tasks demand onboard sensing instead of privileged information (*e.g.*, ground-truth object positions and navigation map). All the tasks use the GeometricGoal (Batra et al., 2020) specification $(s_0, s_*)$, which describes the initial 3D (center-of-mass) position $s_0$ of the target object and the goal position $s_*$. For example, *TidyHouse* is specified by 5 tuples $\{(s_0^i, s_*^i)\}_{i=1...5}$.

### 3.2 SUBTASK AND SKILL

In this section, we present the definition of subtask and skill in the context of reinforcement learning. A long-horizon task can be formulated as a Markov decision process (MDP) [1] defined by a tuple $(\mathcal{S}, \mathcal{A}, R, P, \mathcal{I})$ of state space $\mathcal{S}$, action space $\mathcal{A}$, reward function $R(s, a, s')$, transition distribution $P(s'|s, a)$, initial state distribution $\mathcal{I}$. A subtask $\omega$ is a smaller MDP $(\mathcal{S}, \mathcal{A}_\omega, R_\omega, P, \mathcal{I}_\omega)$ derived from the original MDP of the full task. A skill (or policy), which maps a state $s \in \mathcal{S}$ to an action $a \in \mathcal{A}$, is learned for each subtask by RL algorithms.

Szot et al. (2021) introduces several parameterized skills for the HAB: *Pick, Place, Open fridge, Close fridge, Open drawer, Close drawer, Navigate*. Each skill takes a 3D position as input, either $s_0$ or $s_*$. See Appendix C for more details. Here, we provide a brief description of these skills.

**Pick**($s_0$): pick the object initialized at $s_0$
**Place**($s_*$): place the held object at $s_*$
**Open [container]**($s$): open the container containing the object initialized at $s$ or the goal position $s$
**Close [container]**($s$): close the container containing the object initialized at $s$ or the goal position $s$
**Navigate**($s$): navigate to the start of other skills specified by $s$

Note that $s_0$ is constant per episode instead of a tracked object position. Hence, the target object may not be located at $s_0$ at the beginning of a skill, *e.g.*, picking an object from an opened drawer. Next, we will illustrate how these skills are chained in the HAB.

### 3.3 SKILL CHAINING

Given a task decomposition, a hierarchical approach also needs to generate high-level actions to select a subtask and perform the corresponding skill. Task planning (Fikes & Nilsson, 1971) can be applied to find a sequence of subtasks before execution, with perfect knowledge of the environment. An alternative is to learn high-level actions through hierarchical RL. In this work, we use the subtask sequences generated by a perfect task planner (Szot et al., 2021). Here we list these sequences, to highlight the difficulty of tasks [2].

---

[1] To be precise, the tasks studied in this work are partially observable Markov decision process (POMDP).
[2] We only list the subtask sequence of *TidyHouse* for one object here for illustration. The containers are denoted with subscripts $fr$ (fridge) and $dr$ (drawer) if included in the skill.

**TidyHouse**$(s_0^i, s_*^i)$: Navigate$(s_0^i)$ → Pick$(s_0^i)$ → Navigate$(s_*^i)$ → Place$(s_*^i)$

**PrepareGroceries**$(s_0^1, s_*^1, s_0^2, s_*^2, s_0^3, s_*^3)$: Navigate$_{\text{fr}}(s_0^1)$ → Pick$_{\text{fr}}(s_0^1)$ → Navigate$(s_*^1)$ → Place$(s_*^1)$ → Navigate$_{\text{fr}}(s_0^2)$ → Pick$_{\text{fr}}(s_0^2)$ → Navigate$(s_*^2)$ → Place$(s_*^2)$ → Navigate$(s_0^3)$ → Pick$(s_0^3)$ → Navigate$_{\text{fr}}(s_*^3)$ → Place$_{\text{fr}}(s_*^3)$

**SetTable**$(s_0^1, s_*^1, s_0^2, s_*^2)$: Navigate$_{\text{dr}}(s_0^1)$ → Open$_{\text{dr}}(s_0^1)$ → Pick$_{\text{dr}}(s_0^1)$ → Navigate$(s_*^1)$ → Place$(s_*^1)$ → Navigate$_{\text{dr}}(s_0^1)$ → Close$_{\text{dr}}(s_0^1)$ → Navigate$_{\text{fr}}(s_0^2)$ → Open$_{\text{fr}}(s_0^2)$ → Navigate$_{\text{fr}}(s_0^2)$ → Pick$_{\text{fr}}(s_0^2)$ → Navigate$(s_*^2)$ → Place$(s_*^2)$ → Navigate$_{\text{fr}}(s_0^2)$ → Close$_{\text{fr}}(s_0^2)$

# 4 SUBTASK FORMULATION AND SKILL LEARNING FOR MOBILE MANIPULATION

Following the proposed principles (*composability*, *achievability*, *reusability*), we revisit and reformulate subtasks defined in the Home Assistant Benchmark (HAB). The core idea is to enlarge the initial states of manipulation skills to encompass the terminal states of the navigation skill, given our observation that the navigation skill is usually more robust to initial states. However, manipulation skills (*Pick*, *Place*, *Open drawer*, *Close drawer*) in Szot et al. (2021), are stationary. The *composability* of a stationary manipulation skill is restricted, since its feasible initial states are limited due to kinematic constraints. For instance, the robot can not open the drawer if it is too close or too far from the drawer. Therefore, these initial states need to be carefully designed given the trade-off between *composability* and *achievability*, which is not scalable and flexible. On the other hand, the navigation skill, which is learned to navigate to the start of manipulation skills, is also restricted by stationary constraints, since it is required to precisely terminate at a small set of "good" locations for manipulation. To this end, we propose to replace stationary manipulation skills with mobile counterparts. Thanks to mobility, mobile manipulation skills can have better *composability* without sacrificing much *achievability*. For example, a mobile manipulator can learn to first get closer to the target and then manipulate, to compensate for errors from navigation. It indicates that the initial states can be designed in a more flexible way, which also enables us to design a better navigation reward to facilitate learning.

In the context of mobile manipulation, the initial state of a skill consists of the robot base position, base orientation, and joint positions. For simplicity, we do not discuss the initial states of rigid and articulated objects in the scene, which are usually defined in episode generation. Moreover, we follow previous works (Szot et al., 2021; Lee et al., 2021) to initialize the arm at its resting position and reset it after each skill in skill chaining. Such a reset operation is common in robotics (Garrett et al., 2020). Each skill is learned to reset the arm after accomplishing the subtask as in Szot et al. (2021). Furthermore, for base orientation, we follow the heuristic in Szot et al. (2021) to make the robot face the target position $s_0$ or $s_*$.

## 4.1 MANIPULATION SKILLS WITH MOBILITY

We first present how initial base positions are generated in previous works. For stationary manipulation, a feasible base position needs to satisfy several constraints, *e.g.*, kinematic (the target is reachable) and collision-free constraints. Szot et al. (2021) uses heuristics to determine base positions. For *Pick, Place* without containers (fridge and drawer), a navigable position closest to the target position is selected. For *Pick, Place* with containers, a fixed position relative to the container is selected. For *Open, Close*, a navigable position is randomly selected from a handcrafted region relative to each container. Noise is added to base position and orientation in addition, and infeasible initial states are rejected by constraints. See Fig 2 for examples.

The above example indicates the difficulty and complexity to design feasible initial states for stationary manipulation. One naive solution is to enlarge the initial set with infeasible states, but this can hurt learning as shown later in Sec 5.4. Besides, rejection sampling can be quite inefficient in this case, and Szot et al. (2021) actually computes a fixed number of feasible initial states offline.

**Manipulation Skills with Mobility.** To this end, we propose to use mobile manipulation skills instead. The original action space (only arm actions) is augmented with base actions. We devise a unified and efficient pipeline to generate initial base positions. Concretely, we first discretize the floor map with a resolution of $5 \times 5cm^2$, and get all navigable (grid) positions. Then, different candidates are computed from these positions based on subtasks. Candidates are either within a radius (*e.g.*, 2m) around the target position for *Pick, Place*, or a region relative to the container for

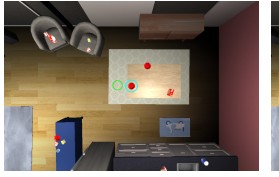 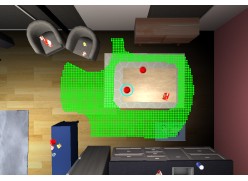 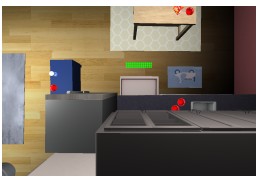 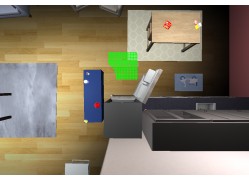

| (a) Pick(stationary) | (b) Pick(mobile) | (c) Close drawer | (d) Close fridge |

Figure 2: Initial base positions of manipulation skills. We only show the examples for *Pick, Close drawer, Close fridge*, as *Place, Open drawer, Open fridge* share the same initial base positions respectively. Positions are visualized as green points on the floor. The target object in *Pick* is highlighted by a circle in cyan. Note that the initial base position of *Pick(stationary)* is a single navigable position closest to the object.

*Open, Close.* Finally, a feasible position is sampled from the candidates with rejection and noise. Compared to stationary manipulation, the rejection rate of our pipeline is much lower, and thus can be efficiently employed on-the-fly during training. See Fig 2 for examples.

### 4.2 NAVIGATION SKILL WITH REGION-GOAL NAVIGATION REWARD

The navigation skill is learned to connect different manipulation skills. Hence, it needs to terminate within the set of initial achievable states of manipulation skills. We follow Szot et al. (2021) to randomly sample a navigable base position and orientation as the initial state of navigation skill. The challenge is how to formulate the reward function, which implicitly defines desirable terminal states. A common navigation reward (Wijmans et al., 2019) is the negative change of geodesic distance to a single 2D goal position on the floor. Szot et al. (2021) extends it for mobile manipulation, which introduces the negative change of angular distance to the desired orientation (facing the target). The resulting reward function, $r_t(s, a)$, for state $s$ and action $a$ is the following (Eq 1):

$$r_t(s, a) = -\Delta_{geo}(g) - \lambda_{ang}\Delta_{ang}I_{[d_t^{geo}(g) \leq \tilde{D}]} + \lambda_{succ}I_{[d_t^{geo}(g) \leq D \wedge d_t^{ang} \leq \Theta]} - r_{slack} \quad (1)$$

$\Delta_{geo}(g) = d_t^{geo}(x_t^{base}, g) - d_{t-1}^{geo}(x_{t-1}^{base}, g)$, where $d_t^{geo}(x_t^{base}, g)$ is the geodesic distance between the current base position $x_t^{base}$ and the 2D goal position $g$. $d_t^{geo}(g)$ is short for $d_t^{geo}(x_t^{base}, g)$. $\Delta_{ang} = d_t^{ang} - d_{t-1}^{ang} = \|\theta_t - \theta^*\|_1 - \|\theta_{t-1} - \theta^*\|_1$, where $\theta_t$ is the current base orientation, and $\theta^*$ is the target orientation. Note that the 2D goal on the floor is different from the 3D goal specification for manipulation subtasks. $I_{[d_t^{geo} \leq \tilde{D}]}$ is an indicator of whether the agent is close enough to the 2D goal, where $\tilde{D}$ is a threshold. $I_{[d_t^{geo} \leq D \wedge d_t^{ang} \leq \Theta]}$ is an indicator of navigation success, where $D$ and $\Theta$ are thresholds for geodesic and angular distances. $r_{slack}$ is a slack penalty. $\lambda_{ang}, \lambda_{succ}$ are hyper-parameters.

This reward has several drawbacks: 1) A single 2D goal needs to be assigned, which should be an initial base position of manipulation skills. It is usually sampled with rejection, as explained in Sec 4.1. It ignores the existence of multiple reasonable goals, introduces ambiguity to the reward (hindering training), and leads the skill to memorize (hurting generalization). 2) There is a hyper-parameter $\tilde{D}$, which defines the region where the angular term $\Delta_{ang}$ is considered. However, it can lead the agent to learn the undesirable behavior of entering the region with a large angular distance, *e.g.*, backing onto the target.

**Region-Goal Navigation Reward.** To this end, we propose a region-goal navigation reward for training the navigation skill. Inspired by object-goal navigation, we use the geodesic distance [3] between the robot and a region of 2D goals on the floor instead of a single goal. Thanks to the flexibility of our mobile manipulation skills, we can simply reuse the candidates (Sec 4.1) for their initial base positions as the navigation goals. However, these candidates are not all collision-free. Thus, we add a collision penalty $r_{col} = \lambda_{col}C_t$ to the reward, where $C_t$ is the current collision force and $\lambda_{col}$ is a weight. Besides, we simply remove the angular term, and find that the success reward is sufficient to encourage correct orientation. Our region-goal navigation reward is as follows:

$$r_t(s, a) = -\Delta_{geo}(\{g\}) + \lambda_{succ}I_{[d_t^{geo}(\{g\}) \leq D \wedge d_t^{ang} \leq \Theta]} - r_{col} - r_{slack} \quad (2)$$

---

[3]The geodesic distance to a region can be approximated by the minimum of all the geodesic distances to grid positions within the region.

# 5 EXPERIMENTS

## 5.1 EXPERIMENTAL SETUP

We use the ReplicaCAD dataset and the Habitat 2.0 simulator (Szot et al., 2021) for our experiments. The ReplicaCAD dataset contains 5 macro variations, with 21 micro variations per macro variation [4]. We hold out 1 macro variation to evaluate the generalization of unseen layouts. For the rest of the 4 macro variations, we split 84 scenes into 64 scenes for training and 20 scenes to evaluate the generalization of unseen configurations (object and goal positions). For each task, we generate 6400 episodes (64 scenes) for training, 100 episodes (20 scenes) to evaluate cross-configuration generalization, and another 100 episodes (the hold-out macro variation) to evaluate cross-layout generalization. The robot is a Fetch (Robotics, 2022) mobile manipulator with a 7-DoF arm and a parallel-jaw gripper. See Appendix B for more details about the setup and dataset generation.

**Observation space:** The observation space includes head and arm depth images ($128 \times 128$), arm joint positions (7-dim), end-effector position (3-dim) in the base frame, goal positions (3-dim) in both base and end-effector frames, as well as a scalar to indicate whether an object is held. The goal position, depending on subtasks, can be either the initial or desired position of the target object. We assume a perfect GPS+Compass sensor and proprioceptive sensors as in Szot et al. (2021), which are used to compute the relative goal positions. For the navigation skill, only the head depth image and the goal position in the base frame are used.

**Action space:** The action space is a 10-dim continuous space, including 2-dim base action (linear forwarding and angular velocities), 7-dim arm action, and 1-dim gripper action. Grasping is abstract as in Batra et al. (2020); Szot et al. (2021); Ehsani et al. (2021). If the gripper action is positive, the object closest to the end-effector within 15cm will be snapped to the gripper; if negative, the gripper will release any object held. For the navigation skill, we use a discrete action space, including a stop action, as in Yokoyama et al. (2021); Szot et al. (2021). A discrete action will be converted to continuous velocities to move the robot, while arm and gripper actions are masked out.

**Hyper-parameters:** We train each skill by the PPO (Schulman et al., 2017) algorithm. The visual observations are encoded by a 3-layer CNN as in Szot et al. (2021). The visual features are concatenated with state observations and previous action, followed by a 1-layer GRU and linear layers to output action and value. Each skill is trained with 3 different seeds. See Appendix C.1 for details.

**Metrics:** Each HAB task consists of a sequence of subtasks to accomplish, as illustrated in Sec 3.3. The completion of a subtask is conditioned on the completion of its preceding subtask. We report progressive completion rates of subtasks, and the completion rate of the last subtask is thus the success rate of the full task. For each evaluation episode, the robot is initialized at a random base position and orientation without collision, and its arm is initialized at the resting position. The completion rate is averaged over 9 different runs [5].

## 5.2 BASELINES

We denote our method by **M3**, short for a multi-skill mobile manipulation pipeline where mobile manipulation skills (**M**) are chained by the navigation skill trained with our region-goal navigation reward (**R**). We compare our method with several RL baselines. All baselines follow the same experimental setup in Sec 5.1 unless specified. We refer readers to Szot et al. (2021) for a sense-plan-act baseline, which is shown to be inferior to the skill chaining pipeline emphasized in this work. Stationary manipulation skills and point-goal navigation reward are denoted by **S** and **P**.

**Monolithic RL (mono)**: This baseline is an end-to-end RL policy trained with a combination of reward functions of individual skills. See Appendix D for more details.

**Stationary manipulation skills + point-goal navigation reward (S+P)**: This baseline is TaskPlanning+SkillsRL (TP+SRL) introduced in Szot et al. (2021), where stationary manipulation skills are chained by the navigation skill trained with the point-goal navigation reward. Compared to the original implementation, we make several improvements, including better reward functions and training schemes. For reference, the original success rates of all HAB tasks are nearly zero. See Appendix A for more details.

---

[4]Each macro variation has a different, semantically plausible layout of large furniture (*e.g.*, kitchen counter and fridge) while each micro variation is generated through perturbing small furniture (*e.g.*, chairs and tables).

[5]3 seeds for RL training multiplied by 3 seeds for initial states

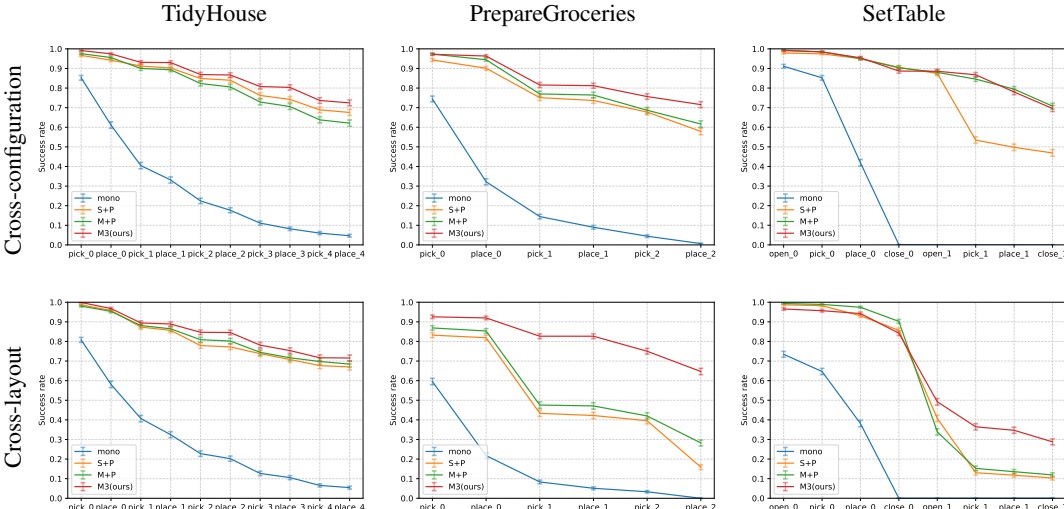

Figure 3: Progressive completion rates for HAB Szot et al. (2021) tasks. The x-axis represents progressive subtasks. The y-axis represents the completion rate of each subtask. The mean and standard error for 100 episodes over 9 seeds are reported. Best viewed zoomed.

**Mobile manipulation skills + point-goal navigation reward (M+P)**: Compared to our **M3**, this baseline does not use the region-goal navigation reward. It demonstrates the effectiveness of proposed mobile manipulation skills. Note that the point-goal navigation reward is designed for the start of stationary manipulation skills.

## 5.3 RESULTS

Fig 3 shows the progressive completion rates of different methods on all tasks. Our method **M3** achieves an average success rate of 71.2% in the cross-configuration setting, and 55.0% in the cross-layout setting, over all 3 tasks. It outperforms all the baselines in both settings, namely **mono** (1.8%/1.8%), **S+P** (57.4%/31.1%) and **M+P** (64.9%/36.2%). First, all the modular approaches show much better performance than the monolithic baseline, which verifies the effectiveness of modular approaches for long-horizon mobile manipulation tasks. Mobile manipulation skills are in general superior to stationary ones (**M+P** *vs.***S+P**). Fig 4 provides an example where mobile manipulation skills can compensate for imperfect navigation. Furthermore, our region-goal navigation reward can reduce the ambiguity of navigation goals to facilitate training (see training curves in Appendix C). Since it does not require the policy to memorize ambiguous goals, the induced skill shows better generalizability, especially in the cross-layout setting (55.0% for **M3** *vs.*36.2% for **M+P**).

## 5.4 ABLATION STUDIES

We conduct several ablation studies to show that mobile manipulation skills are more flexible to formulate than stationary ones, and to understand the advantage of our navigation reward.

**Can initial states be trivially enlarged?** We conduct experiments to understand to what extent we can enlarge the initial states of manipulation skills given the trade-off between *achievability* and *composability*. In the **S(L)+P** experiment, we simply replace the initial states of stationary manipulation skills with those of mobile ones. The success rates of stationary manipulation skills on subtasks drop by a large margin, *e.g.*, from 95% to 45% for *Pick* on *TidyHouse*. Fig 5 shows that **S(L)+P** (37.7%/18.1%) is inferior to both **S+P** (57.4%/31.1%) and **M+P** (64.9%/36.2%). It indicates that stationary manipulation skills have a much smaller set of feasible initial states compared to mobile ones, and including infeasible initial states during training can hurt performance significantly. We also study the impact of initial state distribution on mobile manipulation skills in Appendix F.

**Is the collision penalty important for the navigation skill?** Our region-goal navigation reward benefits from unambiguous region goals and the collision penalty. We add the collision penalty to the point-goal navigation reward (Eq 1) in **S+P(C)** and **M+P(C)** experiments. Fig 5 shows that the collision penalty significantly improves the success rate: **S+P(C)** (65.2%/44.6%) *vs.***S+P** (57.4%/31.1%)

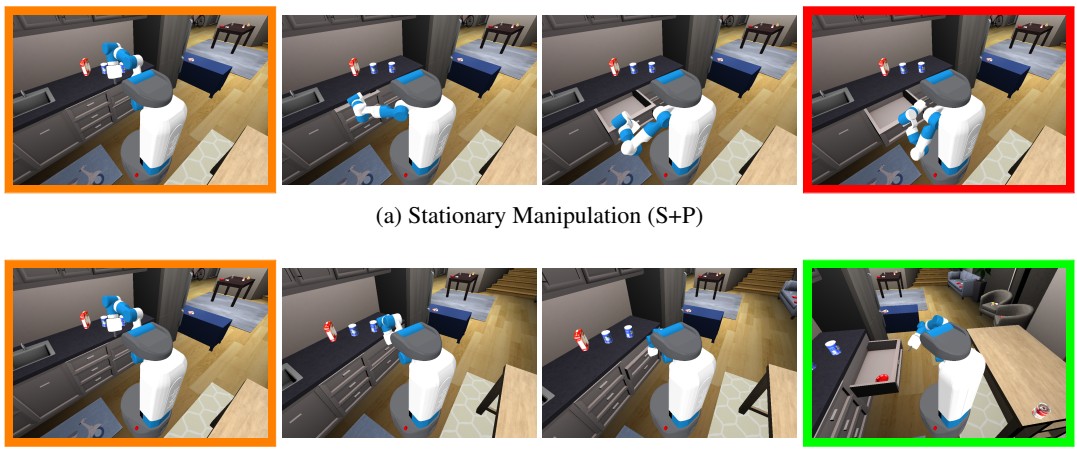

(a) Stationary Manipulation (S+P)

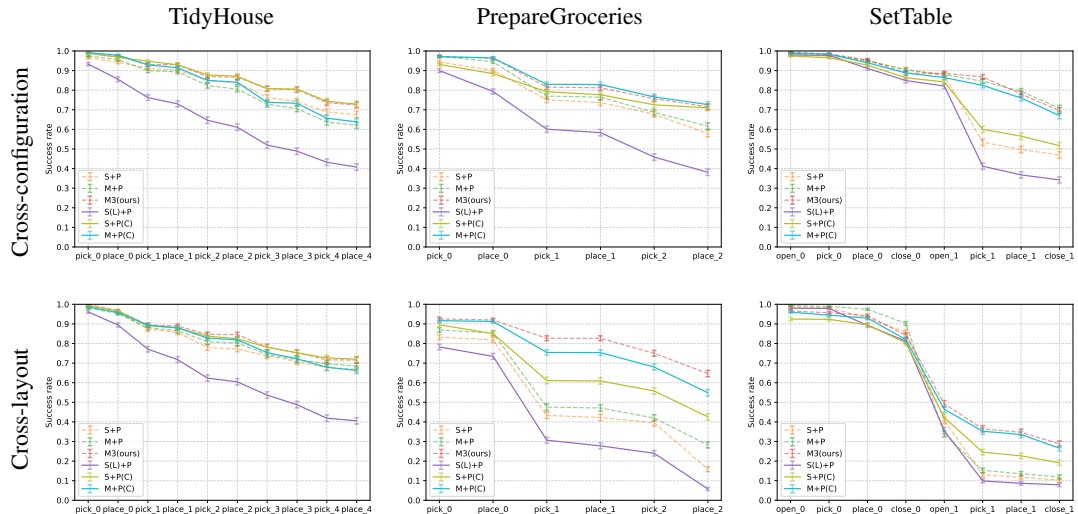

(b) Mobile Manipulation (M+P)

Figure 4: Qualitative comparison between stationary and mobile manipulation. In this example, the point-goal navigation skill terminates between two drawers (1st image). Mobile manipulation manages to open the correct drawer containing the bowl (last image in the bottom row) while stationary manipulation gets confused and finally opens the wrong drawer (last image in the top row). More qualitative results can be found in Appendix H and on our project website.

Figure 5: Progressive completion rates for HAB tasks. The x-axis represents progressive subtasks. The y-axis represents the completion rate of each subtask. Results of ablation experiments are presented with solid lines. The mean and standard error for 100 episodes over 9 seeds are reported.

and **M+P(C)** (67.9%/49.2%) *vs*.**M+P** (64.9%/36.2%). A collision-aware navigation skill can avoid disturbing the environment, *e.g.*, accidentally closing the fridge before placing an object in it. Besides, **M+P(C)** is still inferior to our **M3** (71.2%/55.0%). It implies that reducing the ambiguity of navigation goals helps learn more robust and generalizable navigation skills.

## 6 CONCLUSION AND LIMITATIONS

In this work, we present a modular approach to tackle long-horizon mobile manipulation tasks in the Home Assistant Benchmark (HAB), featuring mobile manipulation skills and the region-goal navigation reward. Given the superior performance, our approach can serve as a strong baseline for future study. Besides, the proposed principles (*achievability, composability, reusability*) can serve as a guideline about how to formulate meaningful and reusable subtasks. However, our work is still limited to abstract grasp and other potential simulation defects. We leave fully dynamic simulation and real-world deployment to future work.

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

## A  OVERVIEW

Compared to the original implementation (Szot et al., 2021), our implementation benefits from re-paired assets (Sec B), improved reward functions and better training schemes (Sec C). Other differences include observation and action spaces. We introduce in observations the target positions in the base frame in addition to those in the end-effector frame. The arm action is defined in the joint configuration space (7-dim) rather than the end-effector Euclidean space (3-dim with no orientation).

## B  DATASET AND EPISODES

Szot et al. (2021) keeps updating the ReplicaCAD dataset. The major fix is "minor furniture layout modifications in order to better accommodate robot access to the full set of receptacles" [6]. The agent radius is also decreased from 0.4m to 0.3m to generate navigation meshes with higher connectivity. Besides, Szot et al. (2021) also improves the episode generator [7] to ensure stable initialization of objects. Those improvements eliminate most unachievable episodes in the initial version. The episodes used in our experiments are generated with the ReplicaCAD v1.4 and the latest habitat-lab [8].

Cross-configuration and cross-layout settings are the same except for scene layouts. In the cross-configuration setting, test scene layouts (micro variations) are different but similar to training ones. In the cross-layout setting, test scene layouts (macro variations) are significantly different from training ones. Each macro variation has a different, semantically plausible layout of large furniture (e.g., kitchen counter and fridge) while each micro variation is generated through perturbing small furniture (e.g., chairs and tables). Thus, the cross-layout setting demands stronger generalization on scene layouts.

For *TidyHouse*, each episode includes 20 clutter objects and 5 target objects along with their goal positions, located at 7 different receptacles (chair, 2 tables, tv stand, two kitchen counters, sofa). For *PrepareGroceries*, each episode includes 21 clutter objects located at 8 different receptacles (the 7 receptacles used in *TidyHouse* and the top shelf of the fridge) and 1 clutter object located at the middle shelf of the fridge. 2 target objects are located at the middle shelf, and each of their goal positions is located at one of two kitchen counters. The third target object is located at one of two kitchen counters, and its goal position is at the middle shelf. *SetTable* generates episodes similar to *PrepareGroceries*, except that two target objects, bowl and apple, are initialized at one of 3 drawers and at the middle fridge shelf respectively. Each of their goal positions is located at one of two tables.

## C  SKILL LEARNING

Each skill is trained to accomplish a subtask and reset its end-effector at the resting position. The robot arm is first initialized with predefined resting joint positions, such that the corresponding resting position of the end-effector is $(0.5, 1.0, 0.0)$ in the base frame [9]. The initial end-effector position is then perturbed by a Gaussian noise $\mathcal{N}(0, 0.025)$ clipped at $0.05m$. The base position is perturbed by a Gaussian noise $\mathcal{N}(0, 0.1)$ truncated at $0.2m$. The base orientation is perturbed by a Gaussian noise $\mathcal{N}(0, 0.25)$ truncated at $0.5$ radian. The maximum episode length is 200 steps for all the manipulation skills, and 500 steps for the navigation skill. The episode terminates on success or failure. We use the same reward function for both stationary and mobile manipulation skills, unless specified.

For all skills, $d_{ee}^o$ is the distance between the end-effector and the object, $d_{ee}^r$ is the distance between the end-effector and the resting position, $d_{ee}^h$ is the distance between the end-effector and a predefined manipulation handle (a 3D position) of the articulated object, $d_a^g$ is the distance between the joint position of the articulated object and the goal joint position. $\Delta_a^b = d_a^b(t-1) - d_a^b(t)$ stands for the (negative) change in distance between $a$ and $b$. For example, $\Delta_{ee}^o$ is the change in distance

---

[6] https://github.com/facebookresearch/habitat-sim/pull/1694
[7] https://github.com/facebookresearch/habitat-lab/pull/764
[8] https://github.com/facebookresearch/habitat-lab/pull/837
[9] The positive x and y axes point forward and upward in Habitat.

between the end-effector and the object. $\mathbb{I}_{holding}$ indicates if the robot is holding an (correct) object or handle. $\mathbb{I}_{succ}$ indicates the task success. $C_t$ refers to the current collision force, and $C_{1:t}$ stands for the accumulated collision force.

The 7-dim arm action stands for the delta joint positions added to the current target joint positions of the PD controller. The input arm action is assumed to be normalized to $[-1, 1]$, and will be scaled by 0.025 (radian). The 2-dim base action stands for linear and angular velocities. The base movement in the Habitat 2.0 is implemented by kinematically setting the robot's base transformation. The collision between the robot base and navigation meshes is taken into consideration. The input base action is assumed to be normalized to $[-1, 1]$, and will be scaled by 3 (navigation skill) or 1.5 (manipulation skills). For the navigation skill, we follow Szot et al. (2021) to use a discrete action space and translate the discrete action into the continuous one. Concretely, the (normalized) linear velocity from -0.5 to 1 is discretized into 4 choices ($\{-0.5, 0, 0.5, 1\}$), and the (normalized) angular velocity from -1 to 1 is discretized into 5 choices ($(\{-1, -0.5, 0, 0.5, 1\})$). The stop action corresponds to the discrete action representing zero velocities.

**Pick($s_0$)**

- Objective: pick the object initialized at $s_0$
- Initial base position (noise is applied in addition):
    - Stationary: the closest navigable position to $s_0$
    - Mobile: a randomly selected navigable position within 2m of $s_0$
- Reward: $\mathbb{I}_{pick}$ indicates whether the correct object is picked and $\mathbb{I}_{wrong}$ indicates whether a wrong object is picked.

$$r_t = 4\Delta_{ee}^o \mathbb{I}_{!holding} + \mathbb{I}_{pick} + 4\Delta_{ee}^r \mathbb{I}_{holding} + 2.5\mathbb{I}_{succ}$$
$$- \max(0.001C_t, 0.2) - \mathbb{I}_{[C_{1:t}>5000]} - \mathbb{I}_{wrong} - \mathbb{I}_{[d_{ee}^o>0.09]}\mathbb{I}_{holding} - 0.002$$

- Success: The robot is holding the target object and the end-effector is within 5cm of the resting position. $\mathbb{I}_{succ} = \mathbb{I}_{holding} \wedge d_{ee}^r \leq 0.05$
- Failure:
    - $\mathbb{I}_{[C_{1:t}>5000]} = 1$: The accumulated collision force is larger than $5000N$.
    - $\mathbb{I}_{wrong} = 1$: A wrong object is picked.
    - $\mathbb{I}_{[d_{ee}^o>0.09]}\mathbb{I}_{holding} = 1$: The held object slides off the gripper.
- Observation space:
    - Depth images from head and arm cameras.
    - The current arm joint positions.
    - The current end-effector position in the base frame.
    - Whether the gripper is holding anything.
    - The starting position $s_0$ in both the base and end-effector frame.
- Action space: The gripper is disabled to release.

**Place($s_*$)**

- Objective: place the held object at $s_*$
- Initial base position (noise is applied in addition):
    - Stationary: the closest navigable position to $s_*$
    - Mobile: a randomly selected navigable position within 2m of $s_*$
- Reward: $\mathbb{I}_{place}$ indicates whether the object is released within 15cm of the goal position, and $\mathbb{I}_{drop}$ indicates whether the object is released beyond 15cm.

$$r_t = 4\Delta_o^{s_*} \mathbb{I}_{holding} + \mathbb{I}_{place} + 4\Delta_{ee}^r \mathbb{I}_{!holding} + 2.5\mathbb{I}_{succ}$$
$$- \min(0.001C_t, 0.2) - \mathbb{I}_{[C_{1:t}>7500]} - \mathbb{I}_{drop} - \mathbb{I}_{[d_{ee}^o>0.09]}\mathbb{I}_{holding} - 0.002$$

- Success: The object is within 15cm of the goal position and the end-effector is within 5cm of the resting position. $\mathbb{I}_{succ} = d_o^{s_*} \leq 0.15 \wedge \mathbb{I}_{!holding} \wedge d_{ee}^r \leq 0.05$

- Failure:
    - $\mathbb{I}_{[C_{1:t}>7500]} = 1$: The accumulated collision force is larger than $7500N$.
    - $\mathbb{I}_{drop} = 1$: The object is released beyond 15cm of the goal position.
    - $\mathbb{I}_{[d_{ee}^o>0.09]}\mathbb{I}_{holding} = 1$: The held object slides off the gripper.
- Observation space:
    - Depth images from head and arm cameras.
    - The current arm joint positions.
    - The current end-effector position in the base frame.
    - Whether the gripper is holding anything.
    - The goal position $s_*$ in both the base and end-effector frame.
- Action space: The gripper is disabled to grasp after releasing the object.

**Open drawer**($s$)

- Objective: open the drawer containing the object initialized at $s$. The goal joint position of the drawer is $g = 0.45m$.
- Initial base position (noise is applied in addition):
    - Stationary: a navigable position randomly selected within a $[0.80, -0.35] \times [0.95, 0.35]$ region in front of the drawer.
    - Mobile: a navigable position randomly selected within a $[0.3, -0.6] \times [1.5, 0.6]$ region in front of the drawer.
- Reward: $\mathbb{I}_{open} = d_a^g \leq 0.05$ indicates whether the drawer is open. $\mathbb{I}_{release}$ indicates whether the handle is released when the drawer is open. $\mathbb{I}_{grasp}$ indicates whether the correct handle is grasped. $a_{base}$ is the (2-dim) base action.

$$r_t = 2\Delta_{ee}^h \mathbb{I}_{!open} + \mathbb{I}_{grasp} + 2\Delta_a^g \mathbb{I}_{holding} + \mathbb{I}_{release} + 2\Delta_{ee}^r \mathbb{I}_{open} + 2.5\mathbb{I}_{succ}$$
$$-\mathbb{I}_{wrong} - \mathbb{I}_{[d_{ee}^h>0.2]}\mathbb{I}_{holding} - \mathbb{I}_{out} - 0.004\|a_{base}\|_1$$

- Success: The drawer is open, and the end-effector is within 15cm of the resting position. $\mathbb{I}_{succ} = \mathbb{I}_{open} \wedge \mathbb{I}_{!holding} \wedge d_{ee}^r \leq 0.15$
- Failure:
    - $\mathbb{I}_{wrong} = 1$: The wrong object or handle is picked.
    - $\mathbb{I}_{[d_{ee}^h>0.2]}\mathbb{I}_{holding} = 1$: The grasped handle slides off the gripper.
    - $\mathbb{I}_{out} = 1$: The robot moves out of a predefined region (a $2m \times 3m$ region in front of the drawer).
    - $\mathbb{I}_{\mathbb{I}_{[open(t-1)\wedge!open(t)]}} = 1$: The drawer is not open after being opened.
    - The gripper releases the handle when the drawer is not open ($\mathbb{I}_{!open} = 1$).
    - $\Delta_a^g >= 0.1$: The drawer is opened too fast.
- Observation space:
    - Depth images from head and arm cameras.
    - The current arm joint positions.
    - The current end-effector position in the base frame.
    - Whether the gripper is holding anything.
    - The starting position $s$ in both the base and end-effector frame.

**Close drawer**($s$)

- Objective: close the drawer containing the object initialized at $s$. The goal joint position is $g = 0m$.
- Initial joint position: $q_a \in [0.4, 0.5]$, where $q_a$ is the joint position of the target drawer. A random subset of other drawers are slightly open ($q_a' \leq 0.1$).
- Initial base position (noise is applied in addition):

- Stationary: a navigable position randomly selected within a $[0.3, -0.35] \times [0.45, 0.35]$ region in front of the drawer.
  - Mobile: a navigable position randomly selected within a $[0.3, -0.6] \times [1.0, 0.6]$ region in front of the drawer.
- Reward: It is almost the same as *Open drawer* by replacing *open* with *close*. $\mathbb{I}_{close} = d_a^g \leq 0.1$.
- Success: The drawer is closed, and the end-effector is within 15cm of the resting position.
- Failure: It is almost the same as *Open drawer* by replacing *open* with *close*, except that the last constraint $\Delta_a^g >= 0.1$ is not included.

**Open fridge($s$)**

- Objective: open the fridge containing the object initialized at $s$. The goal joint position is $g = \frac{\pi}{2}$.
- Initial base position (noise is applied in addition): a navigable position randomly selected within a $[0.933, -1.5] \times [1.833, 1.5]$ region in front of the fridge.
- Reward: $\mathbb{I}_{open} = g - q_a > 0.15$, where $q_a$ is the joint position (radian) of the fridge. To avoid the robot from penetrating the fridge due to simulation defects, we add a collision penalty but excludes collision between the end-effector and the fridge.

$$r_t = 2\Delta_{ee}^h \mathbb{I}_{!open} + \mathbb{I}_{grasp} + +2\Delta_a^g \mathbb{I}_{holding} + \mathbb{I}_{release} + \Delta_{ee}^r \mathbb{I}_{open} + 2.5\mathbb{I}_{succ}$$
$$-\mathbb{I}_{C_{1:t}>5000} - \mathbb{I}_{wrong} - \mathbb{I}_{[d_{ee}^h>0.2]}\mathbb{I}_{holding} - \mathbb{I}_{out} - 0.004\|a_{base}\|_1$$

- Success: The fridge is open, and the end-effector is within 15cm of the resting position. $\mathbb{I}_{succ} = \mathbb{I}_{open} \wedge \mathbb{I}_{!holding} \wedge d_{ee}^r \leq 0.15$
- Failure:
  - $\mathbb{I}_{wrong} = 1$: The wrong object or handle is picked.
  - $\mathbb{I}_{[d_{ee}^h>0.2]}\mathbb{I}_{holding} = 1$: The grasped handle slides off the gripper.
  - $\mathbb{I}_{out} = 1$: The robot moves out of a predefined region (a $2m \times 3.2m$ region in front of the fridge).
  - $\mathbb{I}_{[open(t-1)\wedge!open(t)]} = 1$: The fridge is not open after being opened.
  - The gripper releases the handle when the fridge is not open ($\mathbb{I}_{!open} = 1$).
- Observation space:
  - Depth images from head and arm cameras.
  - The current arm joint positions.
  - The current end-effector position in the base frame.
  - Whether the gripper is holding anything.
  - The starting position $s$ in both the base and end-effector frame.

**Close fridge($s$)**

- Objective: close the fridge containing the object initialized at $s$. The goal joint position is $g = 0$.
- Initial joint position: $q_a \in [\frac{\pi}{2} - 0.15, 2.356]$, where $q_a$ is the joint position of the target fridge.
- Initial base position (noise is applied in addition): a navigable position randomly selected within a $[0.933, -1.5] \times [1.833, 1.5]$ region in front of the fridge.
- Reward: It is almost the same as *Close fridge* by replacing *open* with *close*. $\mathbb{I}_{close} = d_a^g \leq 0.15$.
- Success: The fridge is close, and the end-effector is within 15cm of the resting position.

**Navigate($s$) (point-goal)**

- Objective: navigate to the start of other skills specified by $s$
- Reward: refer to Eq 1. $r_{slack} = 0.002, \tilde{D} = 0.9, \lambda_{ang} = 0.25, \lambda_{succ} = 2.5$
- Success: The robot is within 0.3 meter of the goal, 0.5 radian of the target orientation, and has called the stop action at the current time step.
- Observation space:
  - Depth images from the head camera.
  - The goal position $s_*$ in the base frame.

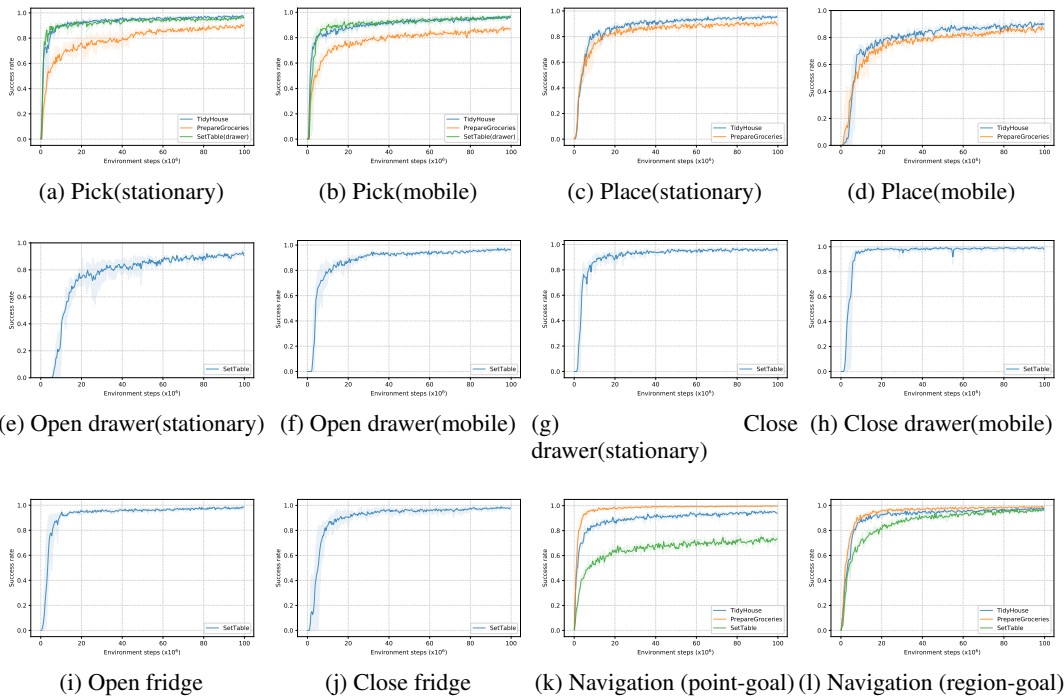

| | | | |
|---|---|---|---|
| (a) Pick(stationary) | (b) Pick(mobile) | (c) Place(stationary) | (d) Place(mobile) |
| (e) Open drawer(stationary) | (f) Open drawer(mobile) | (g) Close drawer(stationary) | (h) Close drawer(mobile) |
| (i) Open fridge | (j) Close fridge | (k) Navigation (point-goal) | (l) Navigation (region-goal) |

Figure 6: Training curves for skills. The y-axis represents the success rate of the subtask (including resetting the end-effector at its resting position). Best viewed zoomed.

**Navigate($s$) (region-goal)**

- Objective: navigate to the start of other skills specified by $s$
- Reward: refer to Eq 2. $r_{slack} = 0.002, r_{col} = \min(0.001C_t, 0.2), \lambda_{succ} = 2.5$
- Success: The robot is within 0.1 meter of any goal in the region, 0.25 radian of the target orientation at the current position, and has called the stop action at the current time step.
- Observation space:
  - Depth images from the head camera.
  - The goal position $s_*$ in the base frame.

### C.1 PPO HYPER-PARAMETERS

Our PPO implementation is based on the habitat-lab. The visual encoder is a simple CNN [10]. The coefficients of value and entropy losses are 0.5 and 0 respectively. We use 64 parallel environments and collect 128 transitions per environment to update the networks. We use 2 mini-batches, 2 epochs per update, and a clipping parameter of 0.2 for both policy and value. The gradient norm is clipped at 0.5. We use the Adam optimizer with a learning rate of 0.0003. The linear learning rate decay is enabled. The mean of the Gaussian action predicted by the policy network is activated by $\tanh$. The (log) standard deviation of the Gaussian action, which is an input-independent parameter, is initialized as $-1.0$. Fig 6 shows training curves of skills.

### C.2 OTHER IMPLEMENTATION DETAILS

The PPO algorithm implemented by the habitat-lab does not distinguish the termination of the environment (MDP) and the truncation due to time limit. We fix this issue in our implementation. Furthermore, we separately train all the skills for each HAB task to avoid potential ambiguity. For example, the starting position of an object in the drawer is computed when the drawer is closed at

---

[10]`https://github.com/facebookresearch/habitat-lab/blob/main/habitat_baselines/rl/models/simple_cnn.py`

the beginning of an episode. However, the skill *Pick* needs to pick this object up when the drawer is open and the actual position of the object is different from the starting position. It is inconsistent with other cases when the object is in an open receptacle or the fridge. We observe such ambiguity can hurt performance. See Fig 6 for all task-specific variants of skills.

## D    Monolithic Baseline

For the monolithic baseline, a monolithic RL policy is trained for each HAB task. During training, the policy only handles one randomly selected target object, *e.g.*, picking and placing one object in *TidyHouse*. During inference, the policy is applied to each target object. We use the same observation space, action space and training scheme as those for our mobile manipulation skills. The main challenge is how to formulate a reward function for those complicated long-horizon HAB tasks that usually require multiple stages. We follow Szot et al. (2021) to composite reward functions for individual skills, given the sequence of subtasks. Concretely, at each time step during training, we infer the current subtask given perfect knowledge of the environment, and use the reward function of the corresponding skill. To ease training, we remove the collision penalty and do not terminate the episode due to collision. Besides, we use the region-goal navigation reward for the navigation subtask. Thanks to our improved reward functions and better training scheme, our monolithic RL baseline is much better than the original implementation in Szot et al. (2021). However, although able to move the object to its goal position, the policy never learns to release the object to complete the subtask *Place* during training. It might be due to exploration difficulty since *Place* is the last subtask in a long sequence and previous subtasks all require the robot not to release. To boost its performance, we force the gripper to release anything held at the end of execution during evaluation.

## E    Evaluation

### E.1    Sequential Skill Chaining

For evaluation, skills are sequentially executed in the order of their corresponding subtasks, as described in Sec 3.3. The main challenge is how to terminate a skill without privileged information. Basically, each skill will be terminated if its execution time exceeds its max episode length (200 steps for manipulation skills and 500 steps for the navigation skill). The termination condition of *Pick* is that an object is held and the end-effector is within 15cm of the resting position, which can be computed based on proprioception only. The gripper is disabled to release for *Pick*. The termination condition of *Place* is that the gripper holds nothing and the end-effector is within 15cm of the resting position. The gripper is disabled to grasp for *Place*. Besides, anything held will be released when *Place* terminates. For *Open* and *Close*, we use a heuristic from Szot et al. (2021): the skill will terminate if the end-effector is within 15cm of the resting position and it has moved at least 30cm away from the resting position during execution. *Navigate* terminates when it calls the stop action. Furthermore, since the manipulation skills only learn to reset its end-effector, we apply an additional operation to reset the whole arm after each skill. This reset operation is achieved by setting predefined joint positions as the target of the robot's PD controller.

### E.2    Progressive Completion Rate

In this section, we describe how progressive completion rates are computed. The evaluation protocol is the same as Szot et al. (2021) (see its Appendix F), and here we phrase it in a way more friendly to readers with little knowledge of task planning and Planning Domain Definition Language (PDDL). To partially evaluate a HAB task, we divide a full task into a sequence of stages (subgoals). For example, *TidyHouse* can be considered to consist of *pick_0*, *place_0*, *pick_1*, *etc.* Each stage can correspond to multiple subtasks. For example, the stage *pick_i* includes *Navigate($s_0^i$)* and *Pick($s_0^i$)*. Thus, to be precise, the completion rate is computed based on stages instead of subtasks. We define a set of predicates to measure whether the goal of a stage is completed. A stage goal is completed if all the predicates associated with it are satisfied. The predicates are listed as follows:

- `holding(target_obj|i)`: The robot is holding the i-th object.
- `at(target_obj_pos|i,target_goal_pos|i)`: The i-th object is within 15cm of its goal position.

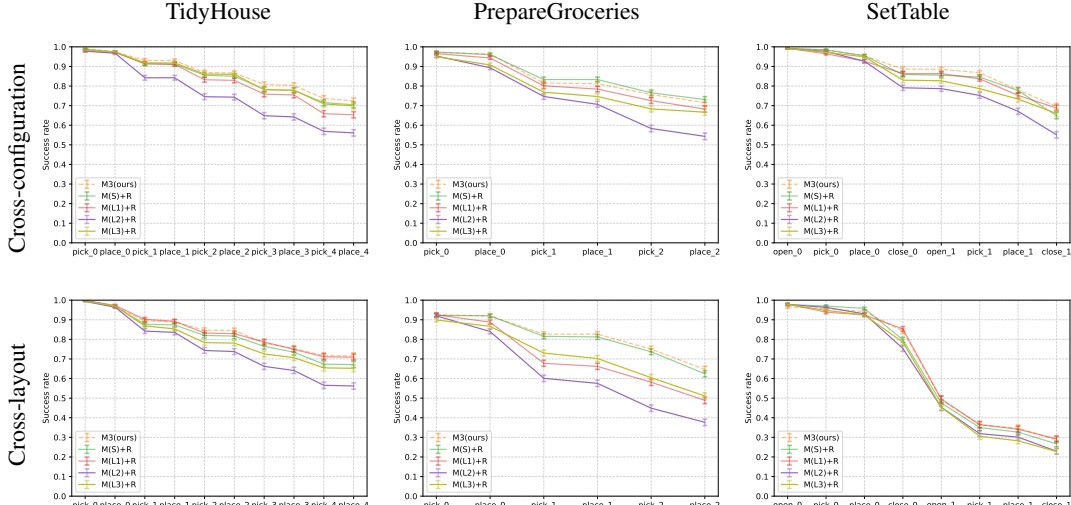

Figure 7: Progressive completion rates for HAB (Szot et al., 2021) tasks. The x-axis represents progressive subtasks. The y-axis represents the completion rate of each subtask. Results of ablation experiments are presented with solid lines. The mean and standard error for 100 episodes over 9 seeds are reported.

- `opened_drawer(target_marker|i)`: The target drawer is open (the joint position is larger than 0.4m).
- `closed_drawer(target_marker|i)`: The target drawer is close (the joint position is smaller than 0.1m).
- `opened_fridge(target_marker|i)`: The target fridge is open (the joint position is larger than $\frac{\pi}{2}$ radian).
- `closed_fridge(target_marker|i)`: The target fridge is close (the joint position is smaller than 0.15 radian).

During evaluation, we evaluate whether the current stage goal is completed at each time step. If the current stage goal is completed, we progress to the next stage. Hence, the completion rate monotonically decreases. Listings 1, 2, 3 present the stages defined for each HAB task and the predicates associated with each stage. Note that the stage goal *place_i* only indicates that the object has been released at its goal position, but the placement can be unstable (*e.g.*, the object falls down the table), which can lead to the failure of the next stage. Besides, due to abstract grasp, it is difficult to place the object stably since the pose of the grasped object can not be fully controlled. Therefore, we modify the objective of *SetTable* to make the task achievable given abstract grasp. Concretely, instead of placing the fruit in the bowl, the robot only needs to place the fruit picked from the fridge at a goal position on the table.

## F    MORE ABLATION STUDIES

In this section, we study the impact of different initial state distributions on mobile manipulation skills. We study the impact of different initial state distributions on mobile manipulation skills. We enlarge initial states by changing the distributions of the initial base position (the radius around the target) and orientation. For reference, the maximum radius around the target is set to 2m in the main experiments (Sec 5). Several experiments are conducted: **M(S)+R, M(L1)+R, M(L2)+R, M(L3)+R**. **M(S)+R**, **M(L1)+R** and **M(L2)+R** stand for the experiments where the maximum radii around the target are set to 1.5m, 2.5m and 4m respectively. **M(L3)+R** keeps the radius as 2m, but samples the initial base orientation from $[-\pi, \pi]$, instead of using the direction facing towards the target. Fig 7 shows the quantitative results. Enlarging the initial states in general leads to performance degradation. Compared to **M3** (71.2%/55.0%), **M(L1)+R** (67.4%/49.7%) and **M(L3)+R**

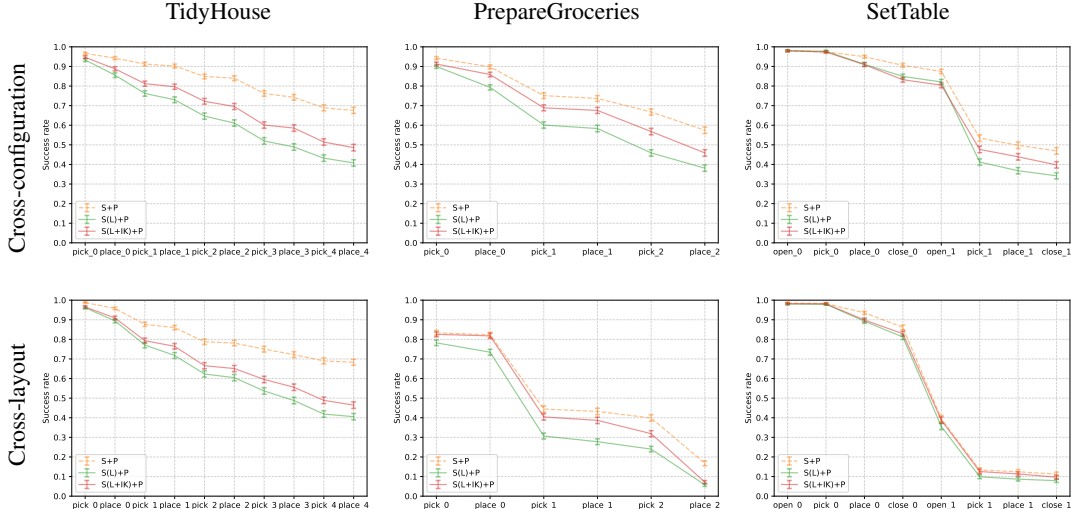

Figure 8: Progressive completion rates for HAB (Szot et al., 2021) tasks. The x-axis represents progressive subtasks. The y-axis represents the completion rate of each subtask. Results of ablation experiments are presented with solid lines. The mean and standard error for 100 episodes over 9 seeds are reported.

(67.5%/46.4%) show moderate performance drop. **M(L2)+R** (55.2%/38.9%) shows the largest performance drop, which indicates that mobile manipulation skills are not able to handle long-range navigation yet. Moreover, **M(S)+R** (69.5%/52.1%) performs on par with **M3**. It implies that there usually exists a "sweet spot" of the initial state distribution for mobile manipulation skills as a trade-off between *achievability* and *composability*.

Besides, we extend the S(L)+P experiment described in Sec 5.4, where we simply replace the initial states of stationary manipulation skills with those of mobile ones. We reject the initial states that the target is not reachable due to the kinematic constraint. The constraint is checked via inverse kinematics (IK). The extended experiment is denoted by **S(L+IK)+P**. Fig 8 shows the quantitative results. The overall success rate of **S(L+IK)+P** is 44.7%/21.1% in the cross-configuration/cross-layout setting. It indicates that increasing the feasible initial states help stationary manipulation skills compared to **S(L)+P** (37.7%/18.1%), but still has a large performance drop compared to **S+P** (57.4%/31.1%). One possible reason is that although the target might be IK-reachable, it can be hard to achieve with stationary manipulation skills due to collision with other objects. However, mobile manipulation skills can first navigate to better locations with fewer obstacles in the front.

## G  MORE QUANTITATIVE METRICS

In this section, we present more quantitative metrics in addition to progressive completion rates for the main experiments on 3 HAB tasks. We report the number of successfully placed objects and the average distance between objects and goals in Table 1 and 2. These metrics are analogous to %FIXEDSTRICT and %E in Weihs et al. (2021).

| Method | TidyHouse (5) | PrepareGroceries (3) | SetTable (2) |
|---|---|---|---|
| S+P | 4.58/4.56 | 2.33/1.54 | 1.45/1.03 |
| M+P | 4.45/4.54 | 2.45/1.79 | 1.71/1.08 |
| M3 (ours) | **4.64/4.60** | **2.56/2.44** | **1.71/1.22** |

Table 1: The number of successfully placed objects for HAB tasks. The metrics in the cross-configuration/cross-layout setting are reported. The number of objects to place is shown along with the name of each task.

| Method | TidyHouse | PrepareGroceries | SetTable |
|---|---|---|---|
| S+P | 0.245/0.256 | 0.338/0.982 | **3.506**/6.481 |
| M+P | 1.296/0.821 | 0.302/0.689 | 4.165/**4.641** |
| M3 (ours) | **0.198/0.174** | **0.239/0.337** | 5.208/6.345 |

Table 2: Average distance between objects and goals for HAB tasks. The metrics in the cross-configuration/cross-layout setting are reported. Note that the average distance is sensitive to outliers.

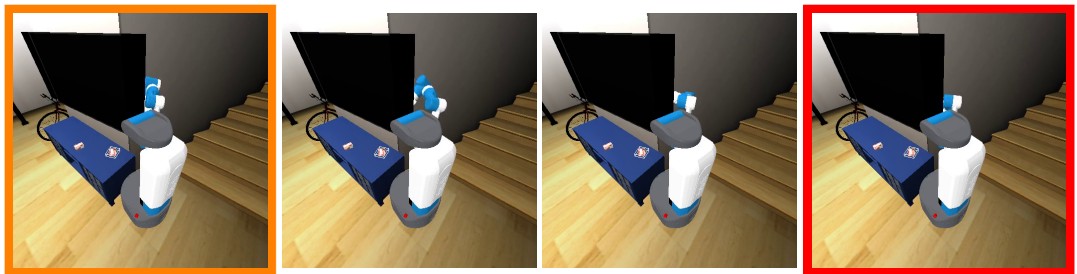

(a) Stationary manipulation and point-goal navigation (S+P)

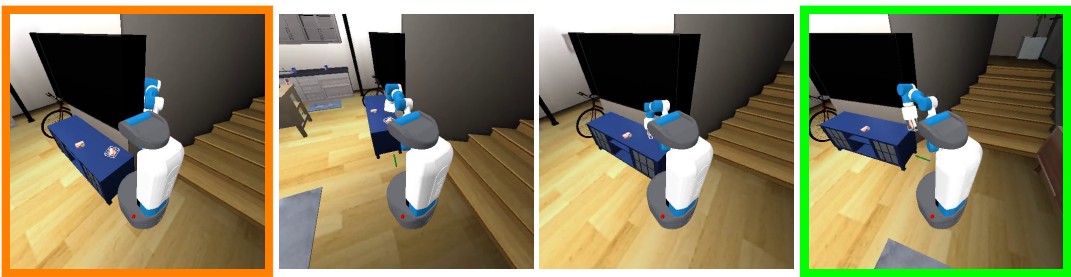

(b) Mobile manipulation and point-goal navigation (M+P)

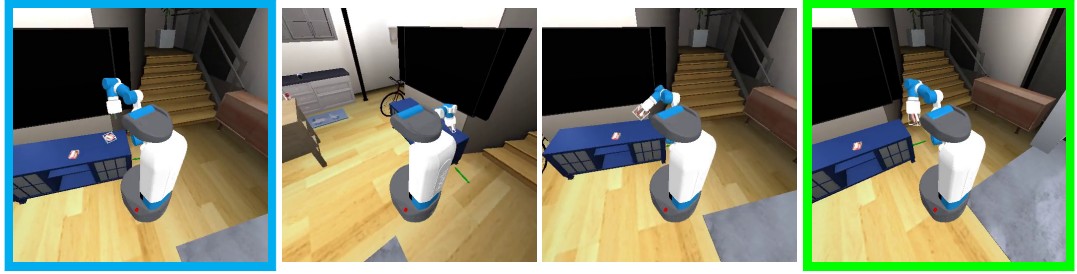

(c) Mobile manipulation and region-goal navigation (M3)

Figure 9: Qualitative comparison in *TidyHouse*. In this example, the point-goal navigation skill terminates behind the TV (1st image). The arm is blocked by the TV in stationary manipulation (last image in the top row). The robot manages to move backward and avoid being blocked in mobile manipulation (last image in the middle row). The region-goal navigation skill instead terminates in front of the TV (1st image in the bottom row).

## H  MORE QUALITATIVE RESULTS

Fig 9, 10, 11 show more qualitative comparison of different methods. Their animated versions can be found on our project website.

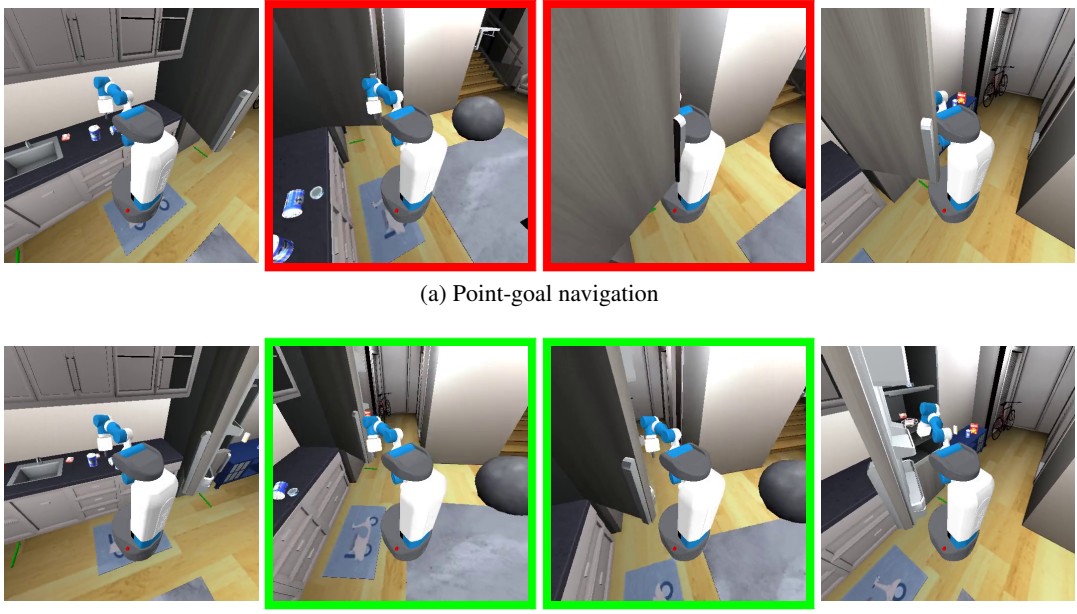

(a) Point-goal navigation

(b) Region-goal navigation

Figure 10: Qualitative comparison in *PrepareGroceries*. In this example, the point-goal navigation skill accidentally close the fridge (top row). The region-goal navigation skill is able to avoid disturbing the environment due to the collision penalty (bottom row).

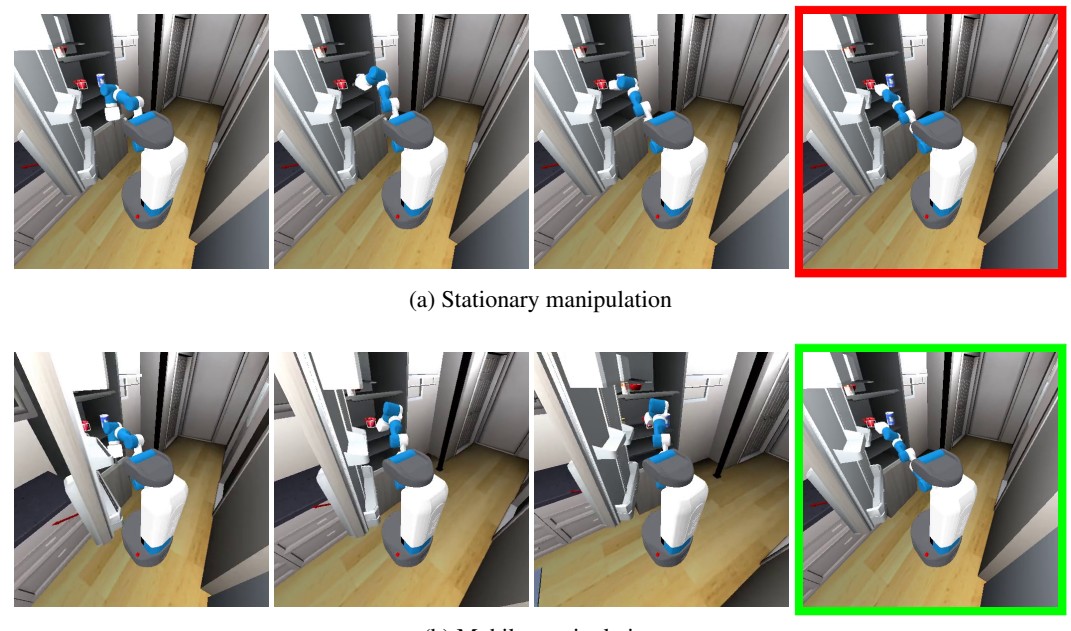

(a) Stationary manipulation

(b) Mobile manipulation

Figure 11: Qualitative comparison in *SetTable*. In this example, the navigation skill terminates at the position where the robot can not reach the target object in the fridge in stationary manipulation. (top row). The robot can move closer to the object and then pick it, to compensate for the navigation skill in mobile manipulation (bottom row).

```yaml
pick_0:
- "holding(target_obj|0)"
place_0:
- "not_holding()"
- "at(target_obj_pos|0,target_goal_pos|0)"
pick_1:
- "holding(target_obj|1)"
- "at(target_obj_pos|0,target_goal_pos|0)"
place_1:
- "not_holding()"
- "at(target_obj_pos|0,target_goal_pos|0)"
- "at(target_obj_pos|1,target_goal_pos|1)"
pick_2:
- "holding(target_obj|2)"
- "at(target_obj_pos|0,target_goal_pos|0)"
- "at(target_obj_pos|1,target_goal_pos|1)"
place_2:
- "not_holding()"
- "at(target_obj_pos|0,target_goal_pos|0)"
- "at(target_obj_pos|1,target_goal_pos|1)"
- "at(target_obj_pos|2,target_goal_pos|2)"
pick_3:
- "holding(target_obj|3)"
- "at(target_obj_pos|0,target_goal_pos|0)"
- "at(target_obj_pos|1,target_goal_pos|1)"
- "at(target_obj_pos|2,target_goal_pos|2)"
place_3:
- "not_holding()"
- "at(target_obj_pos|0,target_goal_pos|0)"
- "at(target_obj_pos|1,target_goal_pos|1)"
- "at(target_obj_pos|2,target_goal_pos|2)"
- "at(target_obj_pos|3,target_goal_pos|3)"
pick_4:
- "holding(target_obj|4)"
- "at(target_obj_pos|0,target_goal_pos|0)"
- "at(target_obj_pos|1,target_goal_pos|1)"
- "at(target_obj_pos|2,target_goal_pos|2)"
- "at(target_obj_pos|3,target_goal_pos|3)"
place_4:
- "not_holding()"
- "at(target_obj_pos|0,target_goal_pos|0)"
- "at(target_obj_pos|1,target_goal_pos|1)"
- "at(target_obj_pos|2,target_goal_pos|2)"
- "at(target_obj_pos|3,target_goal_pos|3)"
- "at(target_obj_pos|4,target_goal_pos|4)"
```

Listing 1: Stage goals and their associated predicates defined for *TidyHouse*. The stages are listed in the order for progressive evaluation.

```
1  pick_0:
2  - "holding(target_obj|0)"
3  place_0:
4  - "not_holding()"
5  - "at(target_obj_pos|0,target_goal_pos|0)"
6  pick_1:
7  - "holding(target_obj|1)"
8  - "at(target_obj_pos|0,target_goal_pos|0)"
9  place_1:
10 - "not_holding()"
11 - "at(target_obj_pos|0,target_goal_pos|0)"
12 - "at(target_obj_pos|1,target_goal_pos|1)"
13 pick_2:
14 - "holding(target_obj|2)"
15 - "at(target_obj_pos|0,target_goal_pos|0)"
16 - "at(target_obj_pos|1,target_goal_pos|1)"
17 place_2:
18 - "not_holding()"
19 - "at(target_obj_pos|0,target_goal_pos|0)"
20 - "at(target_obj_pos|1,target_goal_pos|1)"
21 - "at(target_obj_pos|2,target_goal_pos|2)"
```

Listing 2: Stage goals and their associated predicates defined for *PrepareGroceries*. The stages are listed in the order for progressive evaluation.

```
1   open_0:
2   - "opened_drawer(target_marker|0)"
3   pick_0:
4   - "holding(target_obj|0)"
5   place_0:
6   - "not_holding()"
7   - "at(target_obj_pos|0,target_goal_pos|0)"
8   close_0:
9   - "closed_drawer(target_marker|0)"
10  - "at(target_obj_pos|0,target_goal_pos|0)"
11  open_1:
12  - "closed_drawer(target_marker|0)"
13  - "at(target_obj_pos|0,target_goal_pos|0)"
14  - "opened_fridge(target_marker|1)"
15  pick_1:
16  - "closed_drawer(target_marker|0)"
17  - "at(target_obj_pos|0,target_goal_pos|0)"
18  - "opened_fridge(target_marker|1)"
19  - "holding(target_obj|1)"
20  place_1:
21  - "closed_drawer(target_marker|0)"
22  - "at(target_obj_pos|0,target_goal_pos|0)"
23  - "not_holding()"
24  - "at(target_obj_pos|1,target_goal_pos|1)"
25  close_1:
26  - "closed_drawer(target_marker|0)"
27  - "at(target_obj_pos|0,target_goal_pos|0)"
28  - "closed_fridge(target_marker|1)"
29  - "at(target_obj_pos|1,target_goal_pos|1)"
```

Listing 3: Stage goals and their associated predicates defined for *SetTable*. The stages are listed in the order for progressive evaluation.

