# OpenReview forum: "Multi-skill Mobile Manipulation for Object Rearrangement"
_ICLR.cc/2023/Conference — ICLR 2023 notable top 25%_

### Official Review · Reviewer_x5CG · 2022-10-24

**Confidence:** 4
**Correctness:** 4
**Technical Novelty And Significance:** 3
**Empirical Novelty And Significance:** 4
**Recommendation:** 8

**Clarity, Quality, Novelty And Reproducibility:**

Clarity: This paper is logically clear in describing the method and experiment.
Quality: The quality of this paper could be improved with more experimental detail and evaluation metrics.
Novelty: The motivation is flexible and interesting.
Reproducibility: This paper is reproducible since experimental configurations are clear.


**Strength And Weaknesses:**

Strength:
1. The paper is logically clear in describing the method and experiment. The figures and video are vivid to show the key ideas.
2. The motivation and idea (try to allow the base to move when the manipulation is undertaken) are flexible and interesting. Three principles to formulate subtasks (achievability, composability, reusability) are reasonable.
3. The experiment results show that the proposed 3M model has improved performance in terms of long-horizon tasks.

Weakness:
1. The experiment settings could be explained in more details, such as cross-configuration and cross-layout in Figure 3, 5 and 7. The current version only simply introduces these concepts in Sec 5.1. It would be more clear if the authors could provide more detailed introduction.
2. The experiment metric is only limited to the success rate. Is it possible to evaluate with more metrics like [1]?
3. The paper simply removes the angular term in navigation reward without illustration or experimental evidence.
[1]	Weihs, Luca, Matt Deitke, Aniruddha Kembhavi and Roozbeh Mottaghi. “Visual Room Rearrangement.” 2021 IEEE/CVF Conference on Computer Vision and Pattern Recognition (CVPR) (2021): 5918-5927.


**Summary Of The Paper:**

To tackle long-horizon tasks such as rearrangement, a practical solution is to decompose it into subtasks and complete them sequentially. Each subtask can be solved by designing or learning a skill. However, there exist compounding errors in skill chaining. To tackle this problem, this paper proposes manipulation skills with mobility, which allow the base to move when the manipulation is undertaken, and devise a region-goal navigation reward for mobile manipulation as navigation is the transition between two manipulations. To evaluate the proposed method, this paper conducts experiments on 3 long-horizon mobile manipulation tasks in the Home Assistant Benchmark, showing superior performance compared to the baselines.

**Summary Of The Review:**

The motivations of this paper (region-goal navigation and mobile manipulation skills) are interesting and practical. But I still wonder more sufficient experiments with more metrics.

---

> ### Author Response · Authors · 2022-11-17
> **Response**
>
> We sincerely thank you for acknowledging the strength of our work, e.g., three principles to formulate subtasks. We address the comments and questions below.
>
> ---
>
> > The experiment settings could be explained in more details, such as cross-configuration and cross-layout in Figure 3, 5 and 7. The current version only simply introduces these concepts in Sec 5.1. It would be more clear if the authors could provide more detailed introduction.
>
> All episodes are generated by the official script provided by Habitat (Appendix B). Cross-configuration and cross-layout settings are the same except for scene layouts. In the cross-configuration setting, test scene layouts (micro variations) are different but similar to training ones. In the cross-layout setting, test scene layouts (macro variations) are significantly different from training ones. Each macro variation has a different, semantically plausible layout of large furniture (e.g., kitchen counter and fridge) while each micro variation is generated through perturbing small furniture (e.g., chairs and tables). Thus, the cross-layout setting demands stronger generalization on scene layouts. We added these details to Appendix B.
>
> ---
>
> > The experiment metric is only limited to the success rate. Is it possible to evaluate with more metrics like [1]?
>
> Following the suggestion, we further report the number of successfully placed objects and the average distance between objects and goals, in addition to progressive completion rates. Please refer to the updated Appendix G.
>
> ---
>
> > The paper simply removes the angular term in navigation reward without illustration or experimental evidence.
>
> Thank you for pointing it out. The motivation to remove the angular term is to simplify the reward function and reduce the number of hyperparameters to be tuned during training. Following the suggestion, we conduct additional experiments on all 3 HAB tasks, where the angular term is included in the region-goal navigation reward for M3. We denote this experiment by M3(ang). The success rates of M3(ang) in cross-configuration and cross-layout settings are 69.7% and 56.6% respectively, which are comparable to 71.2% and 55.0% for M3. It indicates that the angular term is not necessary, and removing it leads to a cleaner and easier solution with fewer hyperparameters.

---

### Official Review · Reviewer_w7DJ · 2022-10-25

**Confidence:** 5
**Correctness:** 4
**Technical Novelty And Significance:** 4
**Empirical Novelty And Significance:** 3
**Recommendation:** 10

**Clarity, Quality, Novelty And Reproducibility:**

The paper is written clearly and easy to follow, with helpful visuals and ample details. The reviewer hopes that the author would release their codebase afterward to further demonstrate reproducibility and for other researchers to build on it.

**Strength And Weaknesses:**

The pipeline is novel and performs surprisingly well on long-horizon rearrangement problems with large observation and action space. The experiments are well-designed. The details in their methods and discussions are very informative for researchers in related fields.

**Summary Of The Paper:**

The authors took on a skill-chaining approach to solving long-horizon rearrangement problems and designed region-goal navigation rewards to reduce errors in skill-chaining. Their pipeline is novel and showed encouraging results on difficult multi-skill mobile manipulation problems.

**Summary Of The Review:**

Overall, it is a strong paper, taking a novel model-free approach to solving the difficult problem of long-horizon rearrangement by mobile robots.

---

> ### Author Response · Authors · 2022-11-17
> **Response**
>
> We sincerely thank you for acknowledging the strength of our work. The code is already available in the updated supplementary material and will be released on GitHub. Also, our method ranks first and doubles the success rate of the baseline in the [public HAB challenge](https://eval.ai/web/challenges/challenge-page/1820/leaderboard/4267). We hope our solution can serve as a strong baseline for future research on long-horizon mobile manipulation tasks.

---

### Official Review · Reviewer_zVvP · 2022-10-25

**Confidence:** 4
**Correctness:** 1
**Technical Novelty And Significance:** 2
**Empirical Novelty And Significance:** 2
**Recommendation:** 6

**Clarity, Quality, Novelty And Reproducibility:**

The manuscript is overall clear. The point that the navigation map is not given and it's heavily partially observable, was not clear until I get to Section 5.1
The evaluation is clear and the ablations are helpful for understanding the source of performance gain.
Comparing to previous work on this problem, the changes are a bit limited, where the overall framework is the same, but with an update in skill definition.

**Strength And Weaknesses:**

Rearrangement can be hard when considering partial observability and various physical constraints. The modular approach makes solving each sub-task easier, while posing challenges to the overall feasibility. The proposed changes in this paper are intuitive and are trying to address such challenges. The reward is carefully designed. The ablations in the experiments are helpful and show clear comparison for each change made.

Questions / Weakness
- There are many hard-coded biases injected. For example, the radius selected around the target position for mobile manipulation. How sensitive is the policy to the selection of these values? From the appendix it seems the radius affects the performance a lot.
- 4.1: ‘Finally, a feasible position is sampled from the candidates with rejection and noise.’ What is the rejection based on?
- ‘In the S(L)+P experiment, we simply replace the initial states of stationary manipulation skills with those of mobile ones. ‘ What percentage of initial states are not within the reachable workspace of the robot arm? If doing a simple check that uses IK to determine whether it’s reachable by the robot, would there still be a large performance drop?
- In Fig. 3, M+P is performing much worse than M3. What is the reason for that? As the mobile manipulation should be able to resolve the infeasibility caused by point-goal navigation. Is it more difficult for the mobile manipulation policy to work on a different layout?
- Committing to a specific task plan without checking the intermediate physical constraints is the reason why static manipulation is failing. The current assumption is that the downstream subtask is always feasible and the subgoal is always achievable. If there are some other constraints, for example, if a re-grasp is needed (i.e. pick-place is not feasible but needs pick-place-pick-place), the subtask will still fail, which has the same underlying problem as static manipulation not able to solve the task because of collision or kinematic constraint. The proposed changes can improve the one on reachability. How would the current framework solve this type of problem in general?


**Summary Of The Paper:**

This paper is working on object rearrangement. Comparing to prior work, the proposed method replaces the static manipulation skill with a mobile manipulation skill, which remedies the situation where the navigation target position makes manipulation infeasible due to physical constraints. A region-goal navigation reward is modified based on point-goal navigation reward to train the navigation skill. Experiments on 3 mobile manipulation tasks in HAB show that the proposed modifications can lead to a higher success rate than baselines in most testing settings.

**Summary Of The Review:**

This paper proposes two changes to previous work on solving the rearrangement task. The proposed changes are reasonable. The evaluations are clear. However, the changes to previous work are not significantly large. I'm mostly on the borderline with a little inclination to accept the paper.

---

> ### Author Response · Authors · 2022-11-17
> **Response (1/2)**
>
> We sincerely thank you for your constructive feedback! We address the comments and questions below.
>
> ---
>
> > There are many hard-coded biases injected. For example, the radius selected around the target position for mobile manipulation. How sensitive is the policy to the selection of these values? From the appendix it seems the radius affects the performance a lot.
>
> The initial state distribution indeed has a significant effect on manipulation skills. For example, if the robot is initialized far away from the target, mobile manipulation skills are required to have strong navigation abilities, while stationary manipulation skills are unable to handle such cases at all. When the radius is moderately changed around 2m (+/- 0.5m), the performance drop of mobile manipulation skills is around 5% (Appendix F), while that of stationary counterparts is around 20% (Sec 5.4). Note that the success metric is quite strict since our tasks are long-horizon. For reference, there is a minor or negligible performance change evaluated on subtasks when the radius for mobile manipulation skills changes moderately. Discovering and optimizing the initial state distribution of manipulation skills in the context of skill chaining can be our future work.
>
> ---
>
> > 4.1: ‘Finally, a feasible position is sampled from the candidates with rejection and noise.’ What is the rejection based on?
>
> The rejection is based on collision-free constraints. Concretely, we first sample a candidate position and compute the orientation facing toward the target. Then, we add Gaussian noise to both the position and orientation. Finally, we will reject the sampled initial state if there is a collision between the robot and the environment.
>
> ---
>
> > ‘In the S(L)+P experiment, we simply replace the initial states of stationary manipulation skills with those of mobile ones. ‘ What percentage of initial states are not within the reachable workspace of the robot arm? If doing a simple check that uses IK to determine whether it’s reachable by the robot, would there still be a large performance drop?
>
> First, we try to estimate the percentage of initial states that are not within the reachable workspace. We report the rejection rate (how many samples are rejected until a feasible is found) on the 100 episodes used for the cross-configuration evaluation. For each episode, we need to sample 2 initial states on average, which means the rejection rate is about 50%.
>
> Following the suggestion, we reject initial states where targets are not IK-reachable on the fly during training. We denote the new experiment by S(L+IK)+P. The overall success rate is 44.7%/21.1% in the cross-configuration/cross-layout setting. Detailed results can be found in the updated Appendix F. It indicates that increasing the feasible initial states help stationary manipulation skills compared to S+P (57.4%/31.1%), but still has a large performance drop compared to S(L)+P (37.7%/18.1%). One possible reason is that although the target might be IK-reachable, it can be hard to achieve with stationary manipulation skills due to collision with other objects. However, mobile manipulation skills can first navigate to better locations with fewer obstacles in the front. It is illustrated in the first “TidyHouse” example on the website.
>
> ---
>
> > In Fig. 3, M+P is performing much worse than M3. What is the reason for that? As the mobile manipulation should be able to resolve the infeasibility caused by point-goal navigation. Is it more difficult for the mobile manipulation policy to work on a different layout?
>
> There are two major reasons.
>
> First, the region-goal navigation reward introduces a collision penalty that not only regularizes terminal states but also encourages entire trajectories to be collision-free. It can avoid the robot from disturbing the environment (e.g., closing the fridge or drawer), which is important in long-horizon mobile manipulation tasks like PrepareGroceries and can not be compensated by mobile manipulation skills. In Sec 5.4 (collision penalty), we compare M+P(C) (67.9%/49.2%) and M+P (64.9%/36.2%). We have also shown an example in the PrepareGroceries section on the [project website](https://sites.google.com/view/hab-mobile-manipulation).
>
> Second, the navigation ability of mobile manipulation skills is not strong enough to compensate for all cases. For instance, it is poor at handling cases that are out-of-distribution or require detours. On the other hand, the point-goal navigation skill sometimes wanders between two point goals due to the ambiguity during training, and can terminate at a location worse than both point goals and an orientation facing away from the target. Besides, from Figure 6 in Appendix, the achievability of the point-goal navigation skill is worse than that of the region-goal counterpart, since ambiguous goals lead to worse convergence. Thus, M+P(C) (67.9%/49.2%) is still inferior to M3 (71.2%/55.0%).

---

> ### Author Response · Authors · 2022-11-17
> **Response (2/2)**
>
> > Committing to a specific task plan without checking the intermediate physical constraints is the reason why static manipulation is failing. The current assumption is that the downstream subtask is always feasible and the subgoal is always achievable. If there are some other constraints, for example, if a re-grasp is needed (i.e. pick-place is not feasible but needs pick-place-pick-place), the subtask will still fail, which has the same underlying problem as static manipulation not able to solve the task because of collision or kinematic constraint. The proposed changes can improve the one on reachability. How would the current framework solve this type of problem in general?
>
> Please correct us in case we do not fully understand your question. We will answer how to handle cases like re-grasp.
>
> Re-grasp is one of the classical cases for task and motion planning (TAMP). The key is to sample some intermediate states to augment symbolic state spaces, e.g., waypoints to place the object. Note that in the context of task planning, the assumption still holds that the *action* (skill) can always achieve its *effect* (goal) if *preconditions* (initial states) are satisfied. Therefore, the skills discussed in our work, which are all goal-conditioned, can serve as basic units for high-level task planning. Note that those intermediate states (sub-goals) need to be feasible (e.g., verified by motion planning) and are usually generated based on domain knowledge (e.g., grasp poses). Our multi-skill framework is compatible with the TAMP framework, as long as intermediate states (sub-goals) can be generated.
>
> We would like to further discuss the desired properties of a practical framework for long-horizon mobile manipulation. TAMP usually assumes perfect knowledge of the environment and can not handle cases with rich dynamics. However, the HAB tasks in our work, like many real-world problems, are partially observed. Therefore, a practical solution is online planning. Besides, since even a powerful skill may still not achieve the sub-goal perfectly, we also need planning with uncertainty. These two techniques are also advanced topics for TAMP. *Do As I Can, Not As I Say* (CoRL 2022) shows a potential framework to leverage large-language models to plan, while multiple skills with high achievability and abilities to estimate skill feasibility are still essential.
>
> ---
>
> > The point that the navigation map is not given and it's heavily partially observable, was not clear until I get to Section 5.1
>
> Thanks for pointing this out. Following the suggestion, we add “All the tasks demand onboard sensing instead of privileged information (e.g., ground-truth object positions and navigation map)” in Sec 3.1 in the revision.

---

### Official Review · Reviewer_d9hA · 2022-11-01

**Confidence:** 4
**Correctness:** 4
**Technical Novelty And Significance:** 2
**Empirical Novelty And Significance:** 2
**Recommendation:** 5

**Clarity, Quality, Novelty And Reproducibility:**

Clarity: as commented before, the writing is excellent and clearly conveys the key ideas.
Quality: this is a typical borderline paper in my batch of assigned papers. While the writing / ideas are reasonably good, the technical significant should be further clarified.
Novelty: the idea of using region-goals is somewhat incremental.
Reproducibility: The appendix provides various technical details for the proposed model, including the choosing of parameters and other implementation details. Normally one can re-implement the base code and achieve roughly some performance of the same level as reported in the paper. There seem no hidden trick in the model. Overall I would regard the re-implementation is non-trivial but achievable.

**Details Of Ethics Concerns:**

There are no ethics concerns.

**Strength And Weaknesses:**

Strength:
- The proposed new solution is reasonably improving the performance of a home assisting model. By using region-goal navigation award, the model has higher tolerance for the errors caused in the stage of navigation.
- The writing is good. I have no complaints about the writing.

Weakness:
- The experiments are superficially treated. It is crucial for the readers to understand which key designs brought the performance elevation. Although two ablation studies (w.r.t. initial states or collision penalty) are provided, the proof is mainly based on comparing the success rates of different settings or methods. No further in-depth analysis is given.
- The technical contribution is quite limited. The modification to conventional pipelines is the inclusion of region-goal (rather than point-goal) reward. Though it is proved to be effectiveness by various experiments, one can hardly claim this represents some significant contribution.

**Summary Of The Paper:**

This submission proposes a modular solution for the task of multi-skill manipulation for object rearrangement. The addressed topic is an emerging one in the domain of embodied AI. The work mainly aims to tackle the issue of inaccurate terminal positions of the navigation skill. To mitigate the errors of positions (e.g., not satisfying some kinematic constraints), the new solution adopt a region-goal navigation reward, instead of point-goal reward in existing methods. The idea has been validated on ReplicaCAS and Habitat simulator. The experimental results are positive.

**Summary Of The Review:**

Embodied AI is an emerging and challenging research task. This work attempts to contribute an improved model for manipulation-based object rearrangement. I have several concerns regarding the current version that makes me hesitate to recommend acceptance.

The technical significance of the proposed method should be clarified. For example, Section 4.2 only briefly describe the proposed region-goal navigation reward, without much convincing explanation of the key choices (such as "remove the angular term" since "find that the success reward is sufficient to encourage correct orientation"). This makes the draft somewhat more like a technical report. The corresponding experiments faithful report the success rates, from which the main claims are drawn. A few qualitative examples (e.g., Figure 4) are provided for better illustration. However, in-depth analysis is still missing. I would strongly suggest to make the sections of introduce, related work and preliminary knowledge more concise, leaving more space for more quantitative studies and ablation studies.

---

> ### Author Response · Authors · 2022-11-17
> **Response**
>
> We sincerely thank you for your constructive feedback! We address the comments and questions below.
>
> ---
>
> > The experiments are superficially treated.
>
> We have provided an in-depth analysis on the [project website](https://sites.google.com/view/hab-mobile-manipulation).
> - In brief, mobile manipulation skills are superior to stationary manipulation skills since they can handle “inferior locations” where the navigation skill terminates. For example, a mobile manipulator can learn first to get closer to the fridge and then pick the target object from the fridge, if it can not reach the object given its initial position and orientation. It is illustrated in the “SetTable” example on the website.
> - Besides, the region-goal navigation reward can prevent the navigation skill from memorizing point goals which can be ambiguous across different scenes. For example, the point-goal navigation skill might terminate at a closet navigable location around the target, which is heuristically selected in prior works and might be even not collision-free (e.g., the robot is blocked by the tv in the “TidyHouse” example on the website). Instead, our region-goal navigation reward along with the collision penalty can encourage the skill to figure out collision-free navigable positions spontaneously, to get rid of previous heuristics to select goals.
> - Last, the collision penalty also plays an important role, which avoids the robot from disturbing the environment (e.g., closing the fridge or drawer, or knocking down target objects during navigation). It is illustrated in the “PrepareGroceries” example on the website.
>
> We believe that animated qualitative examples are more intuitive for readers to understand the benefits of our approach. We have also added some analysis in Appendix H in the updated manuscript.
>
> ---
>
> > The technical contribution is quite limited. The modification to conventional pipelines is the inclusion of region-goal (rather than point-goal) reward. Though it is proved to be effectiveness by various experiments, one can hardly claim this represents some significant contribution.
>
> In addition to the region-goal navigation reward, formulating mobile manipulation skills (especially initial states) in the context of skill chaining is also our technical contribution. Formulating skills for chaining demands a systematic study and a joint design of both manipulation and navigation skills. Thus, we emphasize the effect of different initial state distributions on manipulation skills (Sec 5.4 and Appendix F). Note that the initial states of mobile manipulation skills (Sec 4.1) are jointly designed with the terminal states (navigation goals) of the navigation skill (Sec 4.2). Technically, we also provide an efficient way to sample initial robot base positions in a discrete-to-continuous fashion (Sec 4.1), which can be used on the fly during training. We successfully remove most heuristics (e.g., closest navigable locations as initial positions and point goals, hyperparameters for the angular term in the navigation reward) in prior works, which leads to a clean, scalable, and effective solution.
>
> Overall, our solution can serve as a strong baseline for future research on long-horizon mobile manipulation tasks. Our method ranks first and doubles the success rate of the baseline in the [public HAB benchmark](https://eval.ai/web/challenges/challenge-page/1820/leaderboard/4267). Note that the challenge setting is different from this work so the numbers can not be directly compared with those in this work.
>
> ---
>
> > without much convincing explanation of the key choices (such as "remove the angular term" since "find that the success reward is sufficient to encourage correct orientation")
>
> Thank you for pointing it out. The motivation to remove the angular term is to simplify the reward function and reduce the number of hyperparameters to be tuned during training. Following the suggestion, we conduct additional experiments on all 3 HAB tasks, where the angular term is included in the region-goal navigation reward for M3. We denote this experiment by M3(ang). The success rates of M3(ang) in cross-configuration and cross-layout settings are 69.7% and 56.6% respectively, which are comparable to 71.2% and 55.0% for M3. It indicates that the angular term is not necessary, and removing it leads to a cleaner and easier solution with fewer hyperparameters.

---

### Author Response · Authors · 2022-11-17
**General response to all reviewers**

We sincerely thank all reviewers for their constructive feedback, comments, and suggestions.

We summarize our modifications according to feedback:
- Main paper
  - Sec 3.1: “All the tasks demand onboard sensing instead of privileged information (e.g., ground-truth object positions and navigation map).”
  - Sec 5.1: refer readers to Appendix B for more information about the setup
  - Figure 4: refer readers to Appendix H and the project website for more qualitative results
- Appendix
  - Sec B:  more information about the cross-configuration and cross-layout settings
  - Sec F: additional ablation studies about initial states for stationary manipulation skills
  - Sec G: more quantitative metrics
  - Sec H: more qualitative results and analysis
- Supplementary: anonymized codes

Besides, our method was submitted to the public [Habitat Rearrangement Challenge 2022](https://eval.ai/web/challenges/challenge-page/1820/leaderboard/4267), ranked first and doubled the success rate of baselines. We hope our solution can serve as a strong baseline for future research on long-horizon mobile manipulation tasks.

---

### Decision · Program_Chairs · 2023-01-20

**Decision:**

Accept: notable-top-25%

**Justification For Why Not Higher Score:**

It's still a technical method only.

**Justification For Why Not Lower Score:**

Because it works and generalises well.

**Metareview: Summary, Strengths And Weaknesses:**

The paper proposes a modular approach for multi-skill manipulation to rearrange objects. The paper is well-written, the approach is new, and the results are good.


**Note From Pc:**

if the above contains the word "oral" or "spotlight" please see: "oral" presentation means -> notable-top-5% and "spotlight" means -> notable-top-25%. As stated in our emails, we are disassociating presentation type from AC recommendations